# miR-93 regulates Msk2-mediated chromatin remodelling in diabetic nephropathy

Shawn S. Badal[1,2], Yin Wang[1], Jianyin Long[1], David L. Corcoran[3], Benny H. Chang[4], Luan D. Truong[5], Yashpal S. Kanwar[6], Paul A. Overbeek[4] & Farhad R. Danesh[1,2,7]

How the kidney responds to the metabolic cues from the environment remains a central question in kidney research. This question is particularly relevant to the pathogenesis of diabetic nephropathy (DN) in which evidence suggests that metabolic events in podocytes regulate chromatin structure. Here, we show that miR-93 is a critical metabolic/epigenetic switch in the diabetic milieu linking the metabolic state to chromatin remodelling. Mice with inducible overexpression of a miR-93 transgene exclusively in podocytes exhibit significant improvements in key features of DN. We identify miR-93 as a regulator of nucleosomal dynamics in podocytes. miR-93 has a critical role in chromatin reorganization and progression of DN by modulating its target Msk2, a histone kinase, and its substrate H3S10. These findings implicate a central role for miR-93 in high glucose-induced chromatin remodelling in the kidney, and provide evidence for a previously unrecognized role for Msk2 as a target for DN therapy.

[1] Section of Nephrology, The University of Texas MD Anderson Cancer Center, 1400 Pressler Street, Unit 1468, Houston, Texas 77030, USA. [2] Interdepartmental Graduate Program in Translational Biology and Molecular Medicine, Baylor College of Medicine, One Baylor Plaza, Houston, Texas 77030, USA. [3] Center for Genomic and Computational Biology, Duke University, 101 Science Drive, Durham, North Carolina 27708, USA. [4] Department of Molecular and Cellular Biology, Baylor College of Medicine, One Baylor Plaza, Houston, Texas 77030, USA. [5] Department of Pathology and Molecular Medicine, Houston Methodist Hospital, 6565 Fannin Street, Houston, Texas 77030, USA. [6] Department of Pathology, Feinberg School of Medicine, Northwestern University, 303 E Chicago Avenue, Chicago, Illinois 60611, USA. [7] Department of Pharmacology, Baylor College of Medicine, One Baylor Plaza, Houston, Texas 77030, USA. Correspondence and requests for materials should be addressed to F.R.D. (email: fdanesh@mdanderson.org).

Diabetic nephropathy (DN) is a major microvascular complication of both type 1 and 2 diabetes, and the most common cause of end-stage kidney disease in the United States[1]. Prolonged hyperglycaemia leads to chronic metabolic and haemodynamic changes that result in a myriad of genetic and epigenetic changes, which ultimately set the stage for the progression of DN. However, how metabolic responses in the cytoplasm lead to transcriptional and epigenetic changes in the nucleus in DN is not clear.

MicroRNAs (miRNAs) are short noncoding RNAs that generally function through suppression of their complementary target messenger RNAs (mRNAs) via formation of the effector ribonucleoprotein complex RNA-induced silencing complex (RISC). miRNAs are involved in numerous biological processes in the cell, and have emerged as potential targets in the treatment of a wide variety of disease states, including heart failure, cancer and diabetes[2–5]. Studies have linked miRNAs to several kidney diseases[6–9]; we have previously reported that miR-93, a metabolically regulated miRNA, is differentially downregulated in the kidneys of experimental models of diabetes[10]. However, whether restoration of miR-93 expression in kidneys could have therapeutic implications in DN is unexplored. In the present study, we investigate the effect of miR-93 in DN using both genetic and pharmacological approaches, and explore possible mechanisms of how miR-93 can influence progression of DN. Importantly, we define a unique mechanistic role of miR-93 in DN, whereby metabolically regulated miR-93 serves as a metabolic/epigenetic switch in the regulation of chromatin states in podocytes in the diabetic milieu. In addition, we identify Msk2 (mitogen and stress-activated kinase-2; Rps6ka4) as a target of miR-93 and a novel target for DN therapy.

Msk2 is a member of the RSK (Ribosomal S6 Kinase) family of serine/threonine kinases, and an important kinase for Histone H3 Ser10 phosphorylation (H3S10P)[11]. H3S10 is phosphorylated by a select group of kinases, and its phosphorylation by Msk2 is directly involved in nucleosomal remodelling and global transcriptional activation upon exposure to mitogens and stress signals[12–14]. Although H3S10P facilitates chromatin remodelling[15], the impact and influence of Msk2-mediated H3S10 phosphorylation on chromatin remodelling in podocytes, and whether Msk2/H3S10P contribute to the pathogenesis of DN, is mostly unexplored.

In the current study, we find that changes in miR-93 expression, through modulation of Msk2-dependent H3S10P, can lead to widespread changes in chromatin organization and gene transcription. Furthermore, we demonstrate that targeting Msk2 in vivo could provide a target for prevention of DN progression. Our results support a model in which miR-93 by targeting Msk2, a chromatin modifier, regulates a group of seemingly unrelated as well as functionally related genes, greatly amplifying its downstream effect in DN.

## Results

### Generation of a podocyte-specific inducible miR-93 mouse model.
miR-93 is a signature miRNA under high glucose (HG) conditions, whose expression is reduced ∼twofold in several experimental models of DN (Supplementary Fig. 1)[10]. To elucidate the consequences of restoring miR-93 levels in in vivo, we generated a conditional and inducible mouse model of miR-93 overexpression (floxed miR-93[Tg]) (Fig. 1a; Supplementary Fig. 2a–c). To induce podocyte-specific overexpression of miR-93, we employed mice carrying a tamoxifen-inducible improved Cre recombinase (iCre-ER[T2]) expressed under the regulation of the human NPHS2 (podocin) gene promoter (hereafter referred to as Pod-Cre-ER[T2]) previously generated in our laboratory[16].

Pod-Cre-ER[T2] mice were crossed with mTomato/mGFP (mT/mG) reporter mice to further validate the podocyte-specific activity of our Cre mice[17]. Treatment with tamoxifen-induced efficient recombination with a robust induction of mGFP expression in podocytes in Pod-Cre-ER[T2] mice (Fig. 1b,c; Supplementary Fig. 2d). Crossing floxed miR-93[Tg] mice with Pod-Cre-ER[T2] mice allowed for generation of the desired podocyte-specific miR-93 bigenic mice (Pod-CreER[T2];miR-93[Tg] mice, hereafter referred to as miR-93[PodTg] mice) (Fig. 1d,e). Quantitative reverse transcription polymerase chain reaction (qRT-PCR) analysis of podocytes isolated from miR-93[PodTg] mice following tamoxifen induction revealed a physiologically relevant increase (∼twofold) in miR-93 expression levels (Fig. 1f; Supplementary Fig. 2e,f).

We observed no significant changes in kidney histology or podocyte morphology compared to controls as assessed by Periodic-Acid Schiff's (PAS) staining and transmission electron microscopy (TEM) (Fig. 1g). These mice displayed no functionally significant differences in albumin to creatinine ratio (ACR), body weight, blood pressure or fasting blood glucose compared to non-induced controls (Fig. 1h–k). These findings suggest that conditional overexpression of miR-93 in podocytes does not significantly alter kidney homoeostasis or function in non-diabetic mice.

### miR-93 overexpression confers protection against DN.
To investigate whether targeted overexpression of miR-93 in podocytes rescues some of the key features of the DN phenotype, we crossed miR-93[PodTg] mice with Lepr[db/+] mice, an established model of type 2 diabetes, to generate triple transgenic diabetic Lepr[db/db]-miR-93[PodTg] mice (hereafter referred to as db/db:miR-93[PodTg]) (Fig. 2a). Tamoxifen-induced db/db:miR-93[PodTg] mice exhibited similar body weight, blood glucose and kidney/body weight ratios compared with controls (non-induced db/db:miR-93[PodTg] mice) (Fig. 2b–d). However, induction of miR-93 in db/db:miR-93[PodTg] resulted in a significant protection against albuminuria (Fig. 2e,f), mesangial matrix expansion (Fig. 2g,h) and a significant attenuation in glomerular desmin staining, a marker of podocyte injury (Fig. 2g,i). Furthermore, induction of miR-93 in podocytes of diabetic mice prevented podocyte loss and restored the protein levels of synaptopodin and nephrin compared to controls (Supplementary Fig. 3a–c). TEM micrographs from tamoxifen-induced db/db:miR-93[PodTg] mice revealed significantly attenuated podocyte foot process effacement and improved slit diaphragm morphology (Fig. 2g, rows three and four). Furthermore, quantification revealed a significant reduction in glomerular basement membrane (GBM) thickness, reduced foot process effacement and increased foot process density in induced-db/db:miR-93[PodTg] mice (Supplementary Fig. 3d,e). Scanning electron microscopy (SEM) analysis corroborated these observations (Fig. 2g, row five). Taken together, our results demonstrate that miR-93 overexpression in podocytes confers significant renoprotection in DN.

### Pharmacological delivery of miR-93 attenuates key features of DN.
Because replacement therapy of miRNAs in vivo has recently proven promising for the treatment of cancer[18], we speculated that pharmacological replacement of miR-93 in vivo may also protect against progression of DN. To this aim, modified miR-93 mimics for in vivo use were injected intraperitoneally (i.p.) twice a week to diabetic db/db mice (Fig. 2k). Using confocal microscopy, we first tracked the presence of exogenously administered Dy547-labelled miR-93 mimics in vivo, where miR-93 mimics appeared to be predominantly localized in the tubules, and to some extent in glomeruli (Fig. 2l). There was a ∼threefold

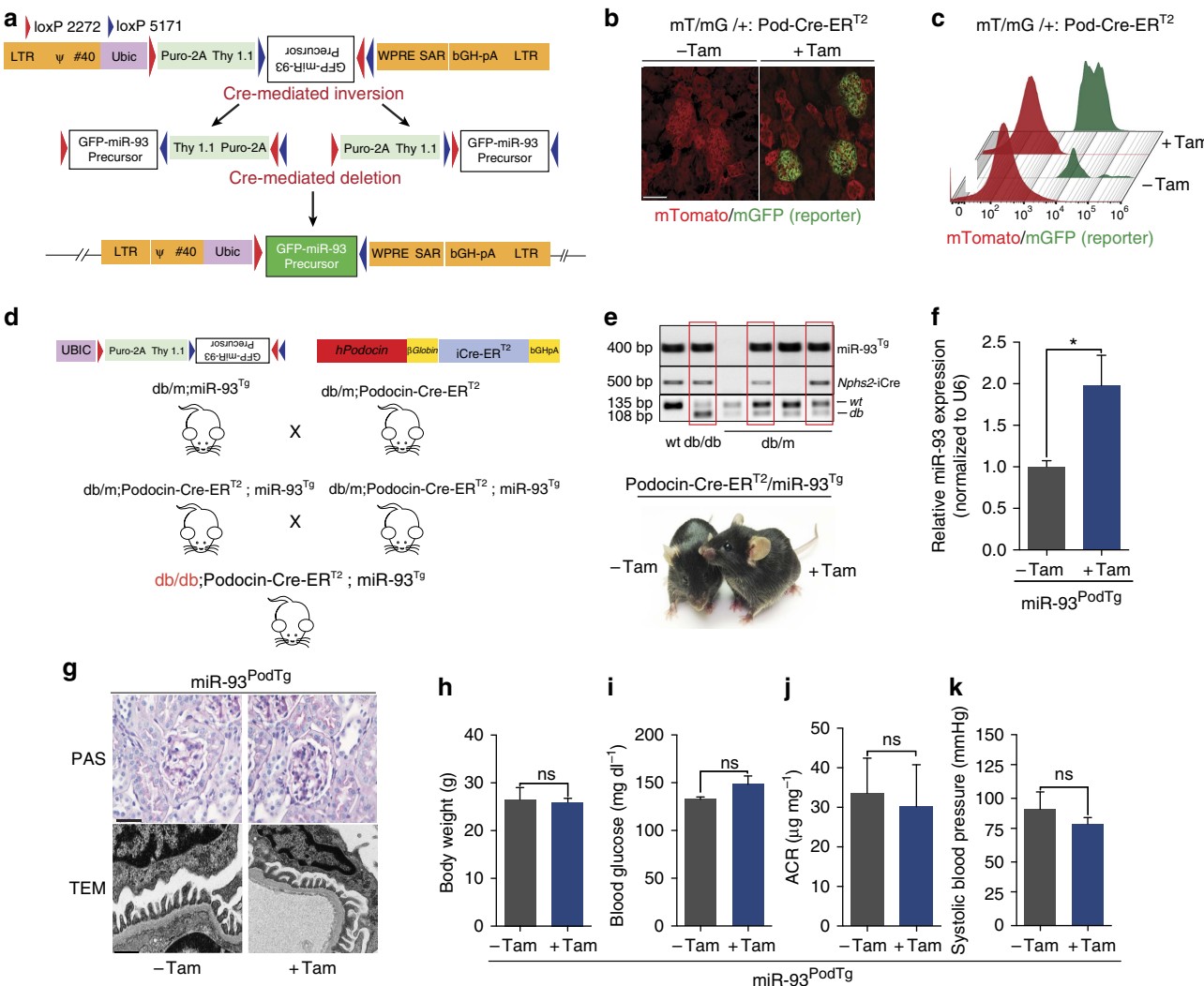

**Figure 1 | Characterization of mice with inducible expression of miR-93 in podocytes. (a)** Schematic design of the LB2-FLIP-GFP-miR-93[Tg] lentivirus used to generate floxed miR-93[Tg] mice. The construct was designed so that Cre induction could be used to mediate the inversion of pre-miR-93 into a sense orientation. Construct elements: LTR: long terminal repeat, Pur: puromycin, Ubic: *Ubiquitin-C* promoter, bGHpA: bovine growth hormone polyadenylation signal; and insulator elements (SAR: scaffold attached regions, WPRE: Woodchuck hepatitis virus post-transcriptional response element, and #40 element). **(b)** Immunofluorescence micrographs of kidney sections collected from control (sesame oil) or tamoxifen-injected mTmG/ + ; *Pod*-CreER[T2] reporter mice. Scale bars, 50 μM. **(c)** Flow cytometry analysis of kidney single cell suspension from control or tamoxifen-injected mTmG/ + ; *Pod*-iCreER[T2] mice (*n* = 3 mice/group). **(d)** Breeding scheme that was used to generate db/db:miR-93[PodTg] mice. **(e)** Top, representative genotyping PCR detecting transgenic miR-93, transgenic Pod-iCreER[T2], and *db* alleles. Boxes highlight representative genotypes from mice carrying the three desired alleles. Bottom, a representative image of *miR-93[PodTg]* mice with or without tamoxifen injections. **(f)** qPCR analysis for miR-93 expression levels in mouse podocytes isolated from control or tamoxifen-injected *miR-93[PodTg]* mice. All values normalized to U6 snRNA internal controls. (*n* = 6 mice/group). **(g)** Top, representative light micrographs of *miR-93[PodTg]* kidney sections stained with PAS from control or tamoxifen-treated mice. Bottom, representative transmission electron micrographs of *miR-93[PodTg]* kidneys from control or tamoxifen-treated mice. Scale bars denote 50 μM and 500 nm, respectively. **(h–k)** Baseline phenotyping of control mice and tamoxifen-injected *miR-93[PodTg]* mice for **(h)** body weight, **(i)** blood glucose, **(j)** ACR and **(k)** systolic blood pressure (*n* = 5 mice/group). Data expressed as mean ± s.e.m. ns: no significance, *$P$ < 0.05. Student's *t*-test was employed for comparisons between two groups.

increase in miR-93 levels in kidneys from miR-93 mimic-treated mice as compared to kidneys from non-targeting (NT) controls (Fig. 2m). Several other organs (for example, liver and spleen) also exhibited robust increases in miR-93 levels compared with mice receiving NT mimics (Fig. 2m). To assess the therapeutic potential of miR-93 in DN, we injected miR-93 mimics or NT mimics to 8-week-old db/db mice twice per week for 8 weeks (Fig. 2k; Supplementary Fig. 3f–g). We found that pharmacological administration of miR-93 reduced proteinuria in diabetic mice (Fig. 2n). Similar to our transgenic approach, we observed significant rescue in several podocyte parameters,

including a significant reduction in glomerular desmin staining, attenuated mesangial matrix expansion, increased WT1 + cells and increased synaptopodin, and nephrin levels in miR-93 mimic-treated mice compared to controls (Fig. 2o, p; Supplementary Fig. 3h–j). Thus, our findings suggest that pharmacological replacement of miR-93 may have therapeutic benefits in DN.

**miR-93 mediates chromatin dynamics in podocytes.** miR-93 does not have its own promoter and is localized in an intron of the *MCM7* gene (Fig. 3a). Since the levels of miR-93 and its host

gene, *MCM7*, in patients with DN are not known, we initially used publicly available datasets from Nephroseq[19] to interrogate the expression level of *MCM7* in patients with DN. We found that *MCM7* mRNA was significantly reduced in patients with DN compared to control subjects (Fig. 3b). We also assessed miR-93 expression levels in human kidney biopsy samples from control subjects and in patients with DN (Supplementary Table 1), and

found miR-93 expression was significantly reduced in patients with DN (∼50%) compared to control subjects (Fig. 3c).

We next interrogated the molecular mechanisms by which miR-93 exerts its protective effects in podocytes. Using standard miRNA target prediction tools, we had previously identified vascular endothelial growth factor (VEGF) as a target of miR-93 *in vitro*[10], and thus we initially validated VEGF as a miR-93 target

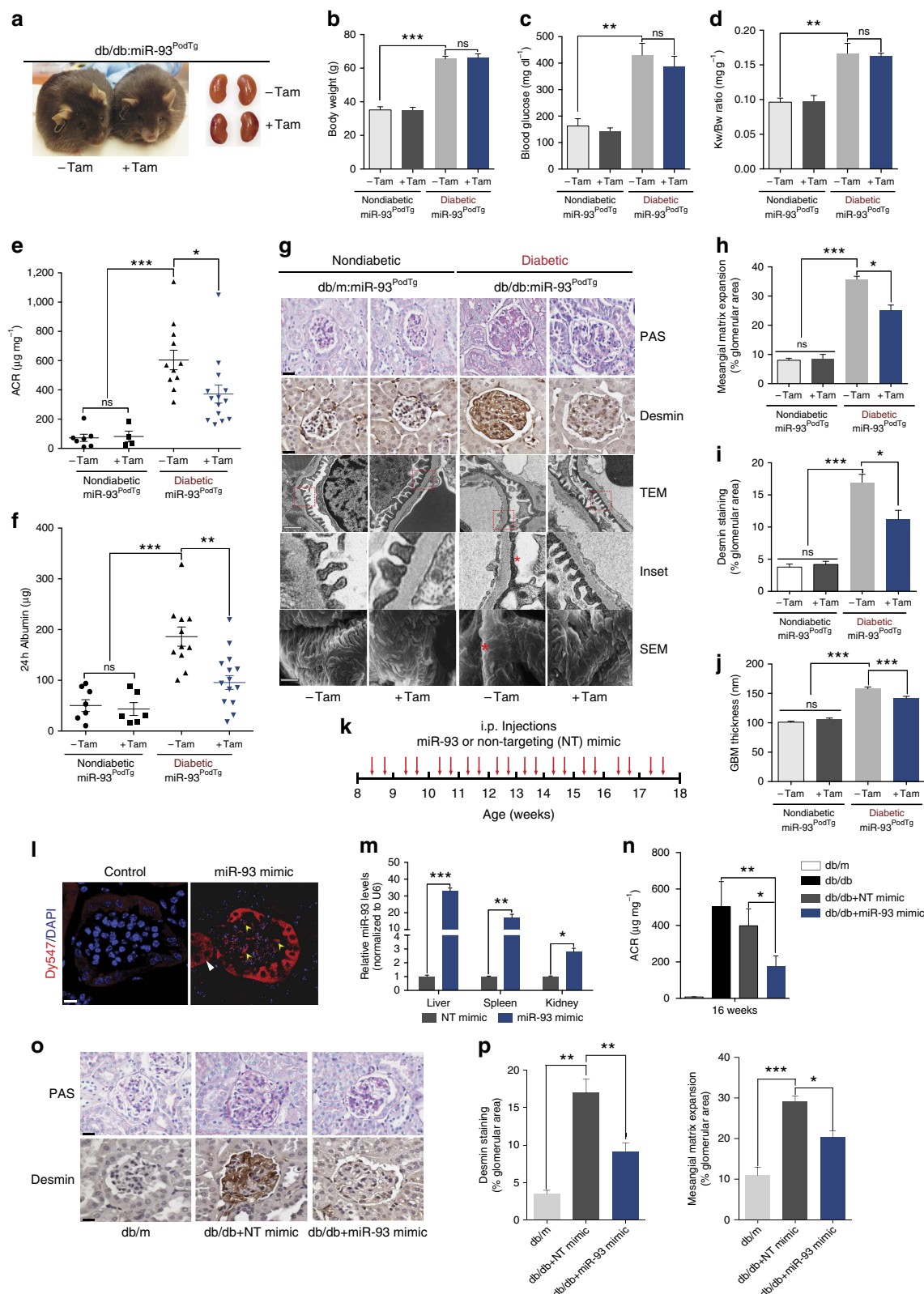

*in vivo* (Supplementary Fig. 4). However, we were interested to use an unbiased approach to identify functionally important miR-93 targets that may be the key to unravelling its biological function in podocytes. Using RNA-Seq analysis, we explored genes differentially regulated by miR-93 overexpression in podocytes (Fig. 3d). The list of significantly downregulated genes was analyzed for the over-representation of any miRNA seed sequence motifs. This analysis confirmed that the miR-93 seed motif was enriched within the 3′-UTRs of significantly downregulated genes (Supplementary Fig. 5a–c). We then searched for differences in the expression of gene sets that may be altered by miR-93 overexpression using the gene set enrichment analysis (GSEA) database[20]. Unexpectedly, GSEA analysis revealed that miR-93 could mediate the expression of genes from several target gene sets implicated in chromatin remodelling (Fig. 3e,f, gene sets highlighted in red).

To gain insight into this previously unrecognized chromatin-modifying function of miR-93 in cultured podocytes in an unbiased manner, we utilized DNaseI hypersensitivity assays coupled with high-throughput sequencing (DNase-Seq) analysis[21–25]. DNase-Seq has evolved into a powerful technique for identifying genome-wide DNase hypersensitive sites (DHS), which typically occur in nucleosome free regions and near gene regulatory elements. We therefore assessed miR-93-modulated DNase1 hypersensitivity changes in podocytes cultured in HG conditions (Fig. 4a). Importantly, we identified a number of HG-related DHS sites, which were reversed upon miR-93 overexpression (Fig. 4b). In particular, our DNase-Seq data revealed that miR-93 overexpression reversed HG-induced changes in hypersensitive sites at the transcription start sites (TSS) of several genes critical in the development of DN, including *Fn1* (fibronectin), *Ctgf* (connective tissue growth factor), *Serpine1* (serpin peptidase inhibitor member 1), *Rock1* (rho-associated, coiled-coil containing protein kinase 1) and *Wt1* (Wilm's tumour protein 1) (Fig. 4c–g)[26,27]. Gain or loss of hypersensitivity upon miR-93 overexpression at these genes coincided with differential patterns of mRNA expression, suggesting changes in chromatin structure were functionally related to changes in gene expression (Fig. 4h; Supplementary Fig. 6). While we observed a number of specific changes in DHS scores at several loci, the location of these sites, as measured by read distribution, was not significantly altered (Supplementary Fig. 7a). This suggests that miR-93 could alter the state of existing DHS sites, but did not necessarily lead to the creation of new sites. Finally, to test whether genes near DHS sites that experienced changes in hypersensitivity were functionally related, we performed GSEA analysis and found miR-93-induced changes in hypersensitive sites significantly affected

pathways important for global regulation of transcription, consistent with the predicted role of miR-93 in chromatin remodelling (Supplementary Fig. 7b). Taken together, these data suggest that one salient but unexpected feature of miR-93 overexpression is its potential to reverse HG-induced changes in chromatin structure and global gene transcription.

In order to tease out the specific downstream target of miR-93 that could mediate this effect, we analyzed the RNA-Seq results and assessed several potential targets of miR-93 with the ability to mediate chromatin remodelling. Among several potential candidates, we chose to test the contribution of the mitogen and stress-activated kinase 2 (Msk2) gene as a differentially regulated target gene (Fig. 4i). Msk2 has previously been shown to regulate serine 10-phosphorylation of histone H3 (H3S10P), an epigenetic mark with an important role in global gene regulation and chromatin dynamics[11,28]. To assess the relevance of Msk2 in DN, we interrogated public datasets in patients with DN, which revealed a significant increase in *MSK2* mRNA expression in DN patients compared to controls (Fig. 4k). Importantly, elevated *MSK2* expression correlated significantly with reduced glomerular filtration rate in the same patients (Fig. 4l). Consistent with our selection, we observed a reduction in RNA tag density at Msk2 in miR-93 mimic transfected podocytes compared to controls (Fig. 4j; Supplementary Fig. 5d), suggesting that high levels of miR-93 are indeed linked to lower Msk2 mRNA levels (Fig. 4j). In addition, podocytes from 24-week-old db/db mice exhibited a significant increase in Msk2 protein expression compared with podocytes obtained from db/m mice (Fig. 4m). Similarly, HG-cultured podocytes demonstrated increased Msk2 expression, compared to NG controls (Fig. 4n,o). Taken together, these findings strongly suggest that Msk2 expression is elevated in experimental models of DN.

**miR-93 mediates Msk2-regulated chromatin remodelling**. We identified a single highly evolutionarily conserved miR-93 binding site within the 3′-UTR of the mouse Msk2 gene, suggesting Msk2 was a bona fide target of miR-93 (Fig. 4p; Supplementary Fig. 5e, left panel). To verify this, we executed several experiments in parallel: first, we performed RNA-IP against endogenous argonaute in cultured podocytes transfected with miR-93 mimics, anti-miR-93 mimics or respective NT controls. *Msk2* mRNA was significantly enriched in RISC complex upon overexpression of miR-93 compared with controls. Inhibition of miR-93 attenuated the incorporation of *Msk2* mRNA into RISC complexes (Fig. 5a). Second, we introduced the full length Msk2 3′-UTR into a luciferase expression vector (Supplementary Fig. 5f), and found that overexpression of miR-93 attenuated reporter activity compared to NT controls. Mutation of the putative miR-93

**Figure 2 | miR-93 overexpression attenuates features associated with DN.** (**a**) Representative images of db/db:*miR-93^PodTg* mice and kidneys from control and tamoxifen-injected groups at 24 weeks of age. (**b–d**) Phenotyping of non-diabetic *miR-93^PodTg* from control ($n = 7$) and tamoxifen-injected mice ($n = 6$) and diabetic *miR-93^PodTg* mice from control ($n = 11$) and tamoxifen-injected ($n = 14$) (**b**) body weight, (**c**) blood glucose and (**d**) kidney weight/body weight (KW/BW) ratio. (**e**) ACR of non-diabetic *miR-93^PodTg* from control ($n = 7$) and tamoxifen-injected mice ($n = 6$) and diabetic *miR-93^PodTg* mice from control ($n = 11$) and tamoxifen-injected ($n = 14$). (**f**) The 24-h albumin excretion rate (AER) from mice as in (**e**). (**g**) Representative images of PAS staining (first row); desmin staining (second row); TEM micrographs (third and fourth rows with magnified inset) and SEM micrographs (fifth row). Red asterisks on TEM and SEM micrographs denote effaced podocyte foot processes. Scale bars denote 50 μM (first row); 200 μM (second row); 0.5 μM (third row); and 1 μM (fourth row). (**h**) Quantification of mesangial matrix expansion, (**i**) glomerular area positive for desmin staining, and (**j**) GBM thickness in non-diabetic and diabetic *miR-93^PodTg* mice from control and tamoxifen-injected groups. ($n = 4–7$ mice/group). (**k**) miRNA mimics injection protocol. (**l**) Fluorescent micrographs of kidney sections from vehicle-injected and 3′-Dy547-miR-93 mimic-injected mice. White arrows specify tubular localization and yellow arrows specify glomerular localization. Scale bars, 50 μM. (**m**) qRT-PCR analysis of liver, spleen and kidney RNAs from NT mimic and miR-93 mimic-injected mice. All values were normalized to U6 snRNA internal controls ($n = 3$ mice/group). (**n**) ACR measurements at 16 weeks of age in listed groups ($n = 8$ mice/group). (**o**) Top, representative PAS staining of kidney sections and bottom, representative desmin staining, respectively of kidney sections from each group. Scale bars denote 50 μM and 200 μM, respectively. (**p**) Quantification of mesangial matrix index and glomerular area positive for desmin staining for each group ($n = 8$ mice/group). Data expressed as mean ± s.e.m. ns: no significance, *$P < 0.05$, **$P < 0.01$. one-way analysis of variance with Tukey's post test for multiple comparisons was used for groups of three or more.

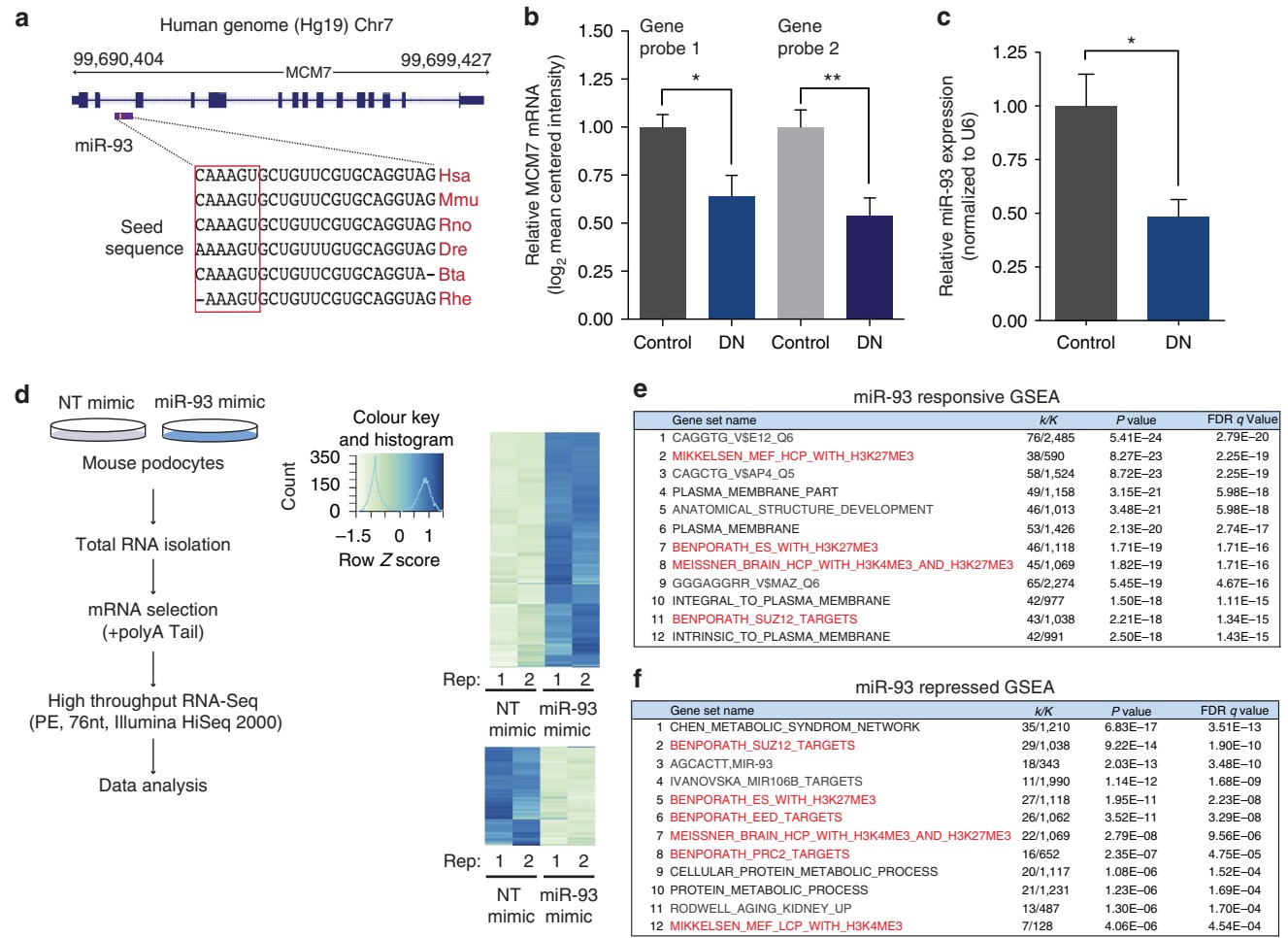

**Figure 3 | RNA-seq analysis implicates miR-93 in chromatin remodelling. (a)** Top, human miR-93 localized to intron 13 of the *MCM7* gene on chromosome 7. Bottom, sequence conservation of the mature miR-93 sequence in several species. Hsa: Homo Sapien, Mmu: *Mus musculus*, Rno: *Rattus norvegicus*, Dre: *Danio rerio*, Bta: *Bos taurus*, Rhe: Rhesus **(b)** *MCM7* expression values in patients with DN for two independent gene array probes obtained from publicly available datasets from Nephroseq (www.nephroseq.org)[19]. Data expressed as $\log_2$ median-centered intensity values. **(c)** miR-93 expression level in biopsy samples from control ($n = 8$) and DN patients ($n = 7$). **(d)** Left, experimental approach employed to generate transcriptome-wide RNA-Seq data. Right, heatmap of differentially regulated genes in NT and miR-93 mimic transfected podocytes. Heatmaps represents hierarchical clustering of the genes, which were significantly differentially regulated ($\log_2$ FC ± 1.0) .**(e,f)** GSEA analysis of **(e)** significantly upregulated and **(f)** significantly downregulated genes upon miR-93 overexpression in mouse podocytes (k/K indicates the number of miR-93 responsive genes over the number of genes in a given gene set). Red highlighted gene sets suggest that miR-93 may play a role in modulating expression of genes regulated by various chromatin remodelers. Data expressed as mean ± s.e.m. ns: no significance, *$P < 0.05$, **$P < 0.01$, ***$P < 0.001$. Student's *t*-test was employed for comparisons between two groups. Cuffdiff2 was used to identify features differentially expressed between conditions for RNA-Seq analysis. A 0.05 false discovery rate was used in selecting significant genes. To further define differentially regulated genes, a cutoff of $-\log_{10}$ ($P$ value) $>1$ and a $\log_2$ fold change (FC) of $<0.5$ or $>0.5$ was employed.

binding site restored luciferase reporter activity to *wild-type* levels (Fig. 5b; Supplementary Fig. 5e, right panel). Third, we analyzed the relationship between miR-93 and Msk2 by western blot analysis, which revealed that miR-93 overexpression, suppressed the translation of Msk2 protein and conversely, inhibition of miR-93 led to a modest increase in Msk2 protein levels (Fig. 5c). Finally, tamoxifen-induced primary cultured podocytes isolated from *miR-93^{PodTg}* mice exhibited a significant reduction in *Msk2* mRNA (Supplementary Fig. 5g). Importantly, HG-induced gain in hypersensitivity and mRNA expression at two well-established Msk2 target genes, *Fos* and *Junb*, were reversed upon miR-93 overexpression (Supplementary Fig. 8).

We then reasoned that one possible explanation for the role of miR-93 on chromatin remodelling could be through the effect of miR-93 on Msk2 and its substrate H3S10P in podocytes, since a growing body of evidence suggests that Msk2 phosphorylation

at H3S10 leads to global chromatin remodelling and gene activation[29,30]. We examined H3S10P in podocytes and found that H3S10 was strongly phosphorylated in cultured podocytes under HG conditions compared with NG conditions (Fig. 5d,e). Furthermore, forced overexpression of miR-93 in HG conditions attenuated Msk2 expression and H3S10P, as measured by western blot and immunofluorescence analysis (Fig. 5d,e). H3S10P is frequently associated with gene-activating acetylation on the same tail (for example, H3K14Ac), as part of triggering an appropriate transcriptional response in cells[31–34]. To test whether the modulatory effect of miR-93 on HG-induced H3S1OP also influences H3 acetylation, we employed a bivalent antibody recognizing H3 tails simultaneously harbouring both H3S10P and H3K14Ac. We found that HG-cultured podocytes treated with miR-93 mimics exhibited a significant reduction in global HG-induced H3S1OP and H3K14Ac (Fig. 5f).

Knockdown of *Msk2* expression was sufficient to recapitulate the effect of miR-93 on H3S10P. We transfected podocytes with siRNA against Msk2 or a scramble siRNA control. Western blot of HG-treated podocytes revealed that siRNA knockdown of Msk2 expression resulted in reduced H3S10P in HG conditions, whereas no striking differences were observed in HG-treated podocytes transfected with a scramble siRNA (Fig. 5g). To assess a link between miR-93 expression and Msk2 *in vivo*, we used our db/db;*miR-93^{PodTg}* mice. We observed that both Msk2 and

H3S10P protein levels in glomeruli were reduced in tamoxifen-induced diabetic mice compared to non-tamoxifen-induced diabetic mice (Fig. 5h,I; Supplementary Fig. 5h). In addition, we observed that db/db mice treated with miR-93-mimics exhibited a significant reduction in glomerular Msk2 and H3S10P levels compared to controls (Fig. 5j,k).

**Targeting of Msk2 *in vivo* prevents progression of DN.** While we have established that Msk2 mediates chromatin remodelling

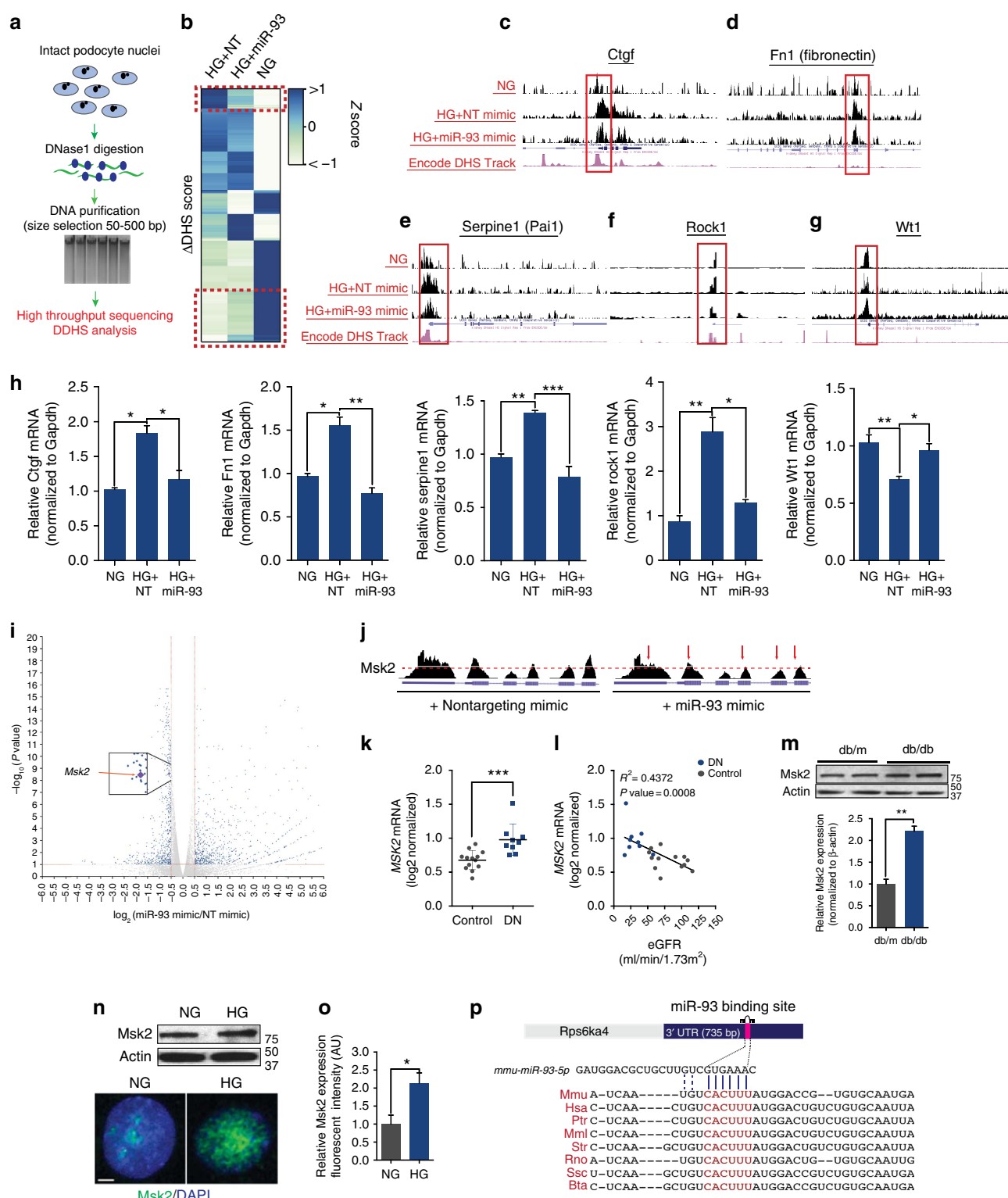

downstream of miR-93, whether targeting Msk2 would be sufficient to prevent the progression of DN *in vivo* remained unknown. To address this, we utilized shRNA plasmids directed against Msk2 coupled with a recently described kidney-specific nanoparticle-based liposomal reagent delivery system[35]. We confirmed knockdown of Msk2 in the podocytes of diabetic mice (Fig. 6a). While we observed no differences in body weight or blood glucose, shMsk2 treatment led to significantly attenuated albuminuria in diabetic (db/db) mice compared to control littermates (Fig. 6b; Supplementary Fig. 9a,b). TEM analysis revealed a significant reduction in GBM thickening and podocyte foot process effacement in shMsk2-treated diabetic mice compared to controls (Fig. 6c,f–h). shMsk2 treatment also prevented podocyte loss (Fig. 6c,e), and resulted in a modest, but significant reduction in mesangial matrix expansion (Fig. 6c,d), we confirmed that H3S10P was reduced in podocytes from shMsk2-treated mice (Fig. 6i). We also found that podocin, nephrin and Wt1 mRNA levels were restored in shMsk2-treated diabetic mice (Fig. 6j; Supplementary Fig. 9c,d). To test if Msk2 depletion by a second method was similarly renoprotective, we employed locked nucleic acids (LNA)-Gapmers directed against Msk2 mRNA (Fig. 6k)[36,37]. We tested the *Msk2* gapmer *in vitro* and observed reduced Msk2 protein expression in podocytes, with no effect on Msk1 expression (Fig. 6l, top panel). To further test the effect of Msk2 gapmer in DN *in vivo*, we employed a similar protocol to our miR-93 mimic injection experiments, wherein 8-week-old db/db mice were treated i.p. with either *Msk2* or a scrambled sequence control gapmer twice a week for 8 weeks (Fig. 6l, bottom panel). Immunoblot analysis of kidney cortices from *Msk2* gapmer administered mice confirmed reduced Msk2 protein expression compared to mice administered scramble gapmer (Fig. 6m). Mice receiving either Msk2 or control gapmer did not exhibit differences in blood glucose (Supplementary Fig. 9e–h), however we observed significantly reduced albuminuria in db/db mice receiving the *Msk2* gapmer compared with db/db mice receiving the scramble gapmer (Fig. 6n). Histological analysis revealed significantly attenuated mesangial matrix expansion and restored podocyte numbers in *Msk2* gapmer-treated mice (Supplementary Fig. 9i–k). Ultrastructure analysis revealed that Msk2 inhibition led to reduced podocyte foot process effacement, reduced foot process width, reduced GBM thickening and increased foot process density (Supplementary Fig. 9l–n). Msk2 gapmer inhibition led to a modest reversal in the expression of several transcript levels related to DN (Supplementary Fig. 9o). Last, while Msk2 knockout mice do not exhibit significant phenotypic changes (Supplementary Fig. 9p–r)[12], diabetic Msk2 knockout mice showed a significant reduction in albuminuria compared to controls (Fig. 6o). Taken together, these data uncover a potential novel contribution of the histone kinase, Msk2, to the pathogenesis of DN.

## Discussion

The molecular mechanisms linking metabolic changes in the cytoplasm to chromatin reorganization in the kidney remain mostly unexplored. In the current study, we elucidated a signalling pathway, whereby metabolically regulated miR-93 has an important role in translating metabolic cues into chromatin reorganization in DN. We first showed that conditional over-expression of miR-93 leads to significant improvements in biochemical and histological features of DN, and pharmacological restoration of miR-93 may have therapeutic implications in DN. Mechanistically, we showed that miR-93 exerts an unexpected role on nucleosomal remodelling via Msk2 and H3S10P in the diabetic milieu. We suggest a model, in which miR-93 serves as a critical metabolic-epigenetic switch in the diabetic environment, linking the metabolic state in the cytoplasm to chromatin reorganization in the nucleus though its target Msk2. In addition, we identified Msk2 as a novel target of miR-93 in DN. Using LNA-modified gapmer oligonucleotides and a kidney-targeted shRNA system, we found that *Msk2* mRNA knockdown leads to significant improvements in key features of DN. Taken together, this study not only unravelled a previously unrecognized miR-93/Msk2/H3S10P signalling pathway, but also helps to introduce and lay the groundwork for future investigation into the role of Msk2 as a novel target for further drug development in patients with DN.

Links between metabolism and nucleosomal remodelling are not unexpected[15]. However, less is known regarding the role of miRNAs in relaying metabolic signalling to alterations in chromatin. In this study, we report a central role of miR-93 in remodelling of the chromatin structure using an unbiased approach. DNase-Seq has evolved into a powerful tool to interrogate global chromatin dynamics in steady state or pathological conditions. In this study, we employed DNase-Seq to assess the impact of miR-93 overexpression on DNase accessibility, and asked whether miR-93 overexpression could alter the chromatin state observed in HG conditions. Alterations of DNase-Seq hypersensitivity were identified at several candidate gene loci provided further evidence that levels of miR-93 are directly linked to the gene expression profiles of several genes critical for the pathogenesis of DN.

Our findings link levels of miR-93 to changes in chromatin structure via H3S10P. This regulatory effect seems to be achieved

**Figure 4 | HG-induced chromatin remodelling is reversed by miR-93 overexpression.** (**a**) Experimental approach employed for DNase-Seq. (**b**) Hierarchical clustering analysis of ΔDHS scores for NG, HG + NT mimic, and HG + miR-93 mimics. Dashed red lines surround regions that demonstrate a reversal in HG-induced changes in hypersensitivity, following miR-93 overexpression. *P* value < 0.05 for all comparisons. (**c–g**) Examples of chromatin accessibility that highlight differences in DNase hypersensitivity at the (**c**) Ctgf, (**d**) Fn1, (**e**) Serpine1 (**f**) Rock1 and (**g**) Wt1 loci. As reference, DHS signal tracks from the Encode project are depicted in pink[24]. Each data track shows tag density from DNase-Seq assays from podocytes treated with NG, HG + NT mimics or HG + miR-93 mimics (**h**) Gene expression analysis of the aforementioned genes. All values normalized to internal Gapdh controls. (**i**) Volcano plot highlighting differentially regulated genes by miR-93. (**j**) mRNA tag density at exons 13–18 and 3′-UTR of murine Msk2 in podocytes transfected with miR-93 mimics or NT controls. (**k**) *MSK2* expression values from patients with DN obtained from publicly available datasets from Nephroseq (nephroseq.org)[19]. Data expressed as $\log_2$ median-centered intensity values. (**l**) Linear regression analysis of relative Msk2 mRNA levels from patients in (**k**) correlated with estimated glomerular filtration rate values obtained from Nephroseq. (**m**) Top, western blot and densitometric quantification of total Msk2 expression levels in podocytes isolated from 24-week-old db/m and db/db mice, ($n = 5$ mice/group). (**n**) Top, western blot of total Msk2 expression levels from whole cell lysates in podocytes cultured under NG or HG conditions. Bottom, immunofluorescence micrographs from podocytes cultured in NG or HG conditions. Scale bar, 10 μm. (**o**) Quantification of nuclear fluorescent intensity of Msk2 expression in podocytes cultured under NG or HG conditions. (**p**) Evolutionary conservation of the miR-93 seed site within the 3′-UTR of Msk2. Mmu: *Mus musculus*, Hsa: *Homo sapiens*, Ptr: *Pan troglodytes*, Mml: *Macaca multta*, Str: *Spermophilus tridecemlineatus*, Rno: *Rattus norvegicus*, Ssc: *Sus scrofa*, Bta: *Bos taurus*. Data expressed as mean ± s.e.m. ns: no significance, *$P < 0.05$, **$P < 0.01$, ***$P < 0.001$. Student's *t*-test was employed for comparisons between two groups, one-way analysis of variance with Tukey's post test for multiple comparisons was used for groups of three or more.

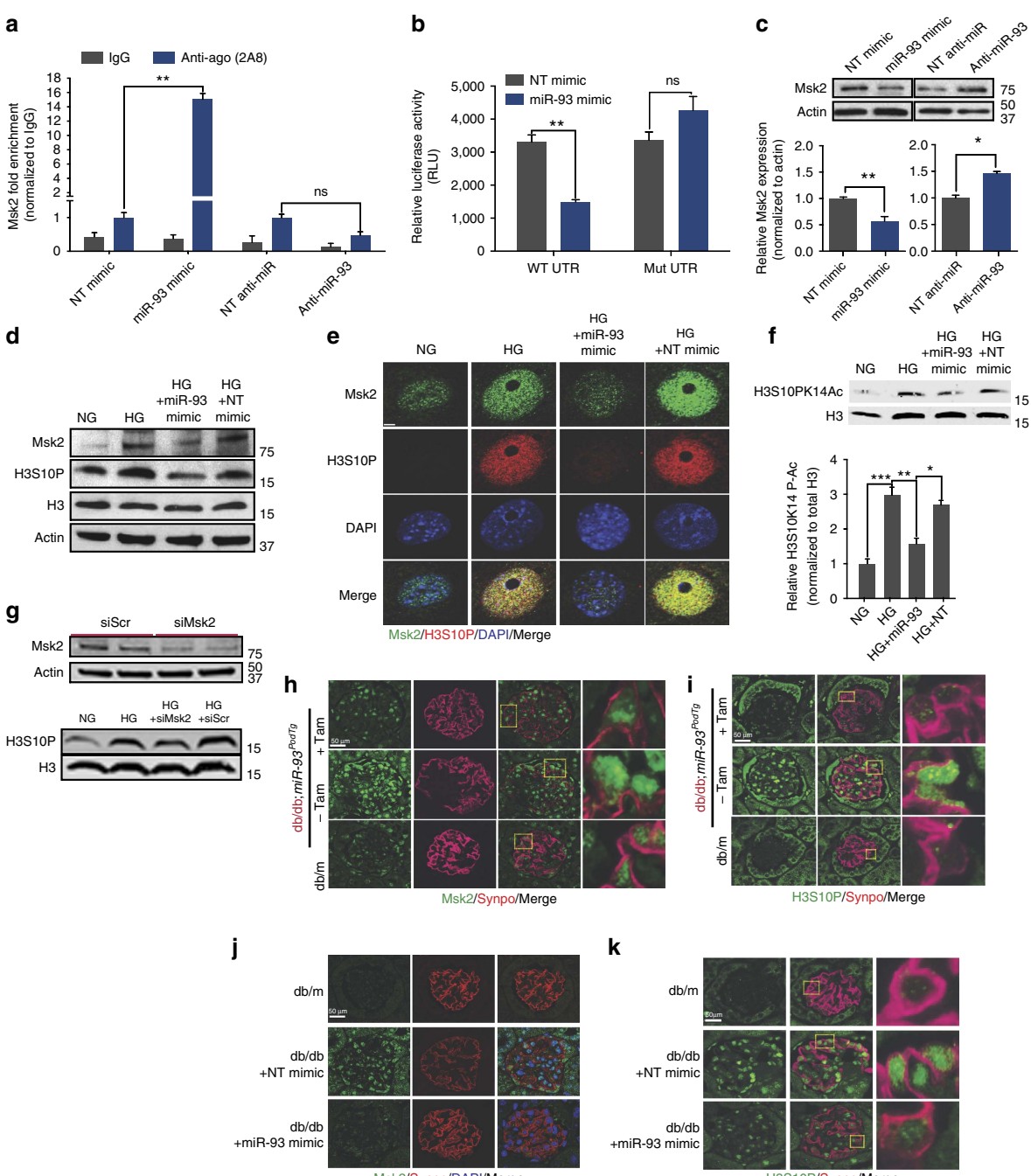

**Figure 5 | miR-93 targets Msk2.** (**a**) Msk2 RNA in RISC complexes from RNA-immunoprecipitation experiments in mouse podocytes transfected with miR-93 mimics or anti-miR-93. (**b**) Luciferase activity in HEK293T cells co-transfected with either 3.1-luc-Msk2 wild-type 3′-UTR (WT-UTR) or a 3.1-luc-Msk2 mutant 3′ UTR (Msk2 mut) and NT or miR-93 mimics, respectively. Luciferase activity levels were normalized to β-gal activities. (**c**) Western blot and corresponding densitometric analysis for total Msk2 expression in podocytes transfected with miR-93 mimics or anti-miR-93 inhibitors. (**d**) Western blot analysis from HG-cultured podocytes transfected with miR-93 mimic or a NT mimic with antibodies directed against Msk2, H3S10P, total H3 and actin. (**e**) Immunofluorescence against H3S10P and Msk2 in HG-cultured podocytes transfected with miR-93 mimic or a NT mimic. Scale bar, 10 μm. (**f**) Western blot analysis of cultured podocytes exposed to HG and transfected with miR-93 mimic or a NT mimic with bivalent antibodies against H3S10K14-P-Ac and total H3. (**g**) Top panel, western blot analysis of Msk2 protein levels in podocytes transfected with a NT siRNA or siRNA specific to Msk2. Bottom panel, western blot analysis of H3S10P and total H3 protein levels in podocytes transfected with siMsk2 or siScr in HG conditions. (**h,i**) Representative immunofluorescence micrographs of kidney sections from db/m (*n* = 3), control db/db;miR-93<sup>PodTg</sup> (*n* = 4) and tamoxifen-induced db/db;miR-93<sup>PodTg</sup> mice (*n* = 4) stained with anti-Synaptopodin and anti-Msk2 or anti-H3S10P antibodies. Sections were counterstained with DAPI. Sections were counterstained with DAPI (**j,k**) Representative immunofluorescence micrographs of kidney sections from db/m control (*n* = 2) and db/db mice allocated to miR-93 mimic (*n* = 3) or NT mimic (*n* = 3) injections with anti-synaptopodin, anti-Msk2 and/or anti-H3S10P antibodies. Merged images are meant to highlight podocyte-specific staining and insets are meant to highlight podocyte-specific localization of Msk2 or H3S10. Sections were counterstained with DAPI. **h–k** Scale bars, 50μm. Data expressed as mean ± s.e.m. ns: no significance, *P < 0.05, **P < 0.01, ***P < 0.001. Student's *t*-test was employed for comparisons between two groups; one-way analysis of variance with Tukey's post test for multiple comparisons was used for groups of three or more.

through Msk2, which serves as a kinase to phosphorylate H3 at Ser10, among other targets[28,31,38]. H3S10P has been shown to have an important role in gene activation following exposure to growth factors or stress signals[31,33,39]. Importantly, Msk2 is the preferential kinase for H3S10 that integrates upstream signals from growth factor and stress signalling to elicit the gene activation response[28,40–42]. One puzzling observation has been

the discovery that H3S10P is also associated with both chromosome condensation and gene activation[40,41]. One possible explanation for this dichotomy could be that the outcome of H3S10 phosphorylation depends on the type of stimulus and the cellular context[15,28,42].

This study provides additional insights into how miRNAs, by modifying chromatin structure, may greatly amplify their gene

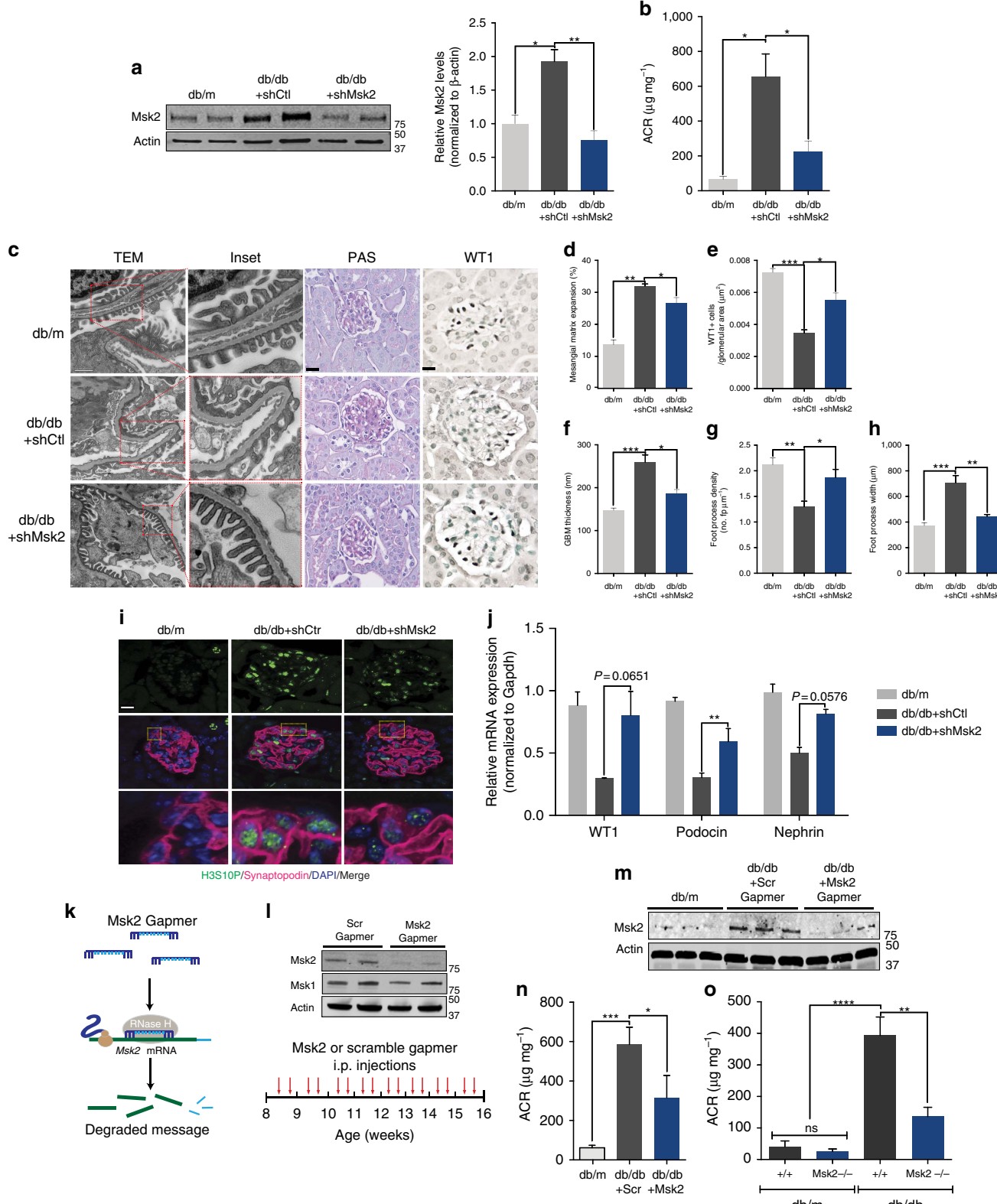

regulatory networks. A prevailing view based largely on *in vitro* studies is that miRNAs exert their effects through 'fine tuning' of numerous functionally related targets[43]. However, our findings provide a view of miRNAs wherein manipulation of a single miRNA, by modifying the chromatin signature, may have a dramatic impact on a disease-related chromatin signature via a broadly acting mechanism. Evidence from our podocyte-specific model of miR-93 overexpression in diabetic mice suggests the miR-93/Msk2/H3S10P cascade in podocytes is critical for progression of DN. However, we recognize that some of the renoprotective effects of targeting this pathway in DN may stem from modulation of chromatin dynamics in other cell types of the kidney, especially since miR-93 and Msk2 expression are both broadly distributed throughout the kidney[10,44]. These assumptions are not entirely surprising as there is considerable cross talk among different cell types of the kidney, and improvements in one cell type might positively impact neighbouring cell types as well[45,46].

Understanding how miRNAs contribute to disease progression is important since miRNA-based therapies for several diseases are currently progressing through early-stage clinical trials[47]. Furthermore, identification of novel miRNA targets may provide unique therapeutic opportunities. Recent preclinical studies have established the role of targeting Msk2 in a variety of diseases[38,48,49]. In this current study, our findings linking global and kidney-specific targeting of Msk2 expression, establish Msk2 as a novel target in DN. Additional work is underway in our laboratory to elucidate the tissue-specific impact of targeting Msk2 in DN.

In conclusion, we propose a previously unrecognized role for miR-93 and Msk2/H3S10P as regulators of nucleosomal remodelling in the diabetic milieu. Furthermore, the findings of this study shed further light on the therapeutic promise of using miR-93 mimics in experimental models of DN, as well as highlight a novel target for therapy by identifying Msk2 as a central mediator of DN progression.

## Methods

**Tissue culture.** Conditionally immortalized mouse podocytes were cultured on BD BioCoat Collagen I plates (BD Biosciences, NJ, USA) at 33 °C in the presence of 20 U ml$^{-1}$ mouse recombinant IFN-γ (Sigma, St Louis, MO, USA) to enhance expression of a thermosensitive T antigen[27]. Mouse podocyte cell lines are routinely characterized in the lab based on morphology and gene expression patterns. Cells for all experiments reported herein were between passage 4–12. Cells were confirmed to be free of mycoplasma contamination. To induce differentiation, podocytes were maintained at 37 °C without IFN-γ for 10–12 days. Podocytes prepared for experiments involving high glucose (HG, 25 mM) conditions were serum deprived for 24 h before addition of HG. Likewise, control cells were serum deprived and cultured with normal glucose, NG, (5 mM). Cell culture experiments were repeated at least three independent times.

**Animal work.** All animal studies were conducted according to the 'Principles of Laboratory Animal Care' (NIH publication No. 85023, revised 1985) and approval was obtained for all animal studies under the guidelines of the IACUC of The University of Texas MD Anderson Cancer Center. Transgenic, Pod-iCreER$^{T2}$ mice were previously generated in our laboratory[16]. To induce Cre recombination, 8-week-old mice with the appropriate genotypes were administered 80 mg kg$^{-1}$ tamoxifen via intraperitoneal injection for 10 days[16,27]. mTom/mGFP mice that express the membrane-targeted tandem dimer tomato (mTdTomato) before Cre-mediated excision and membrane-targeted GFP (mGFP) after excision were obtained from Jackson Laboratories (Bar Harbor, ME, USA). Diabetic db/db mice and their control littermates, db/m, were also obtained from Jackson Laboratories (Strain: BKS.Cg-Dock7$^{m+/+}$Lepr$^{db/J}$, Bar Harbor, ME, USA). Genotyping for mice harbouring the *db* mutation in the *Lepr* gene was carried out using primers specified by Jackson Labs, followed by restriction enzyme digestion with Rsa1. VEGF$^{Hi/+}$ and VEGF$^{lacZ/lacZ}$ were kind gifts from Dr Andras Nagy (Lunenfeld Research Institute, Toronto, ON, Canada). Msk2 Knockout (KO) mice on the C57BL/6J background were a kind gift from Dr Simon Arthur (University of Dundee, Dundee, UK). Homozygous Msk2 KO mice were bred with heterozygous C57BLKS$^{db/m}$ to eventually obtain db/m;Msk2$^{-/-}$. These mice were intercrossed with each other to obtain db/db;Msk2$^{-/-}$. Tail DNA was used to genotype mice at the time of weaning using the REDExtract-n-Amp genotyping kit (Sigma Aldrich, St Louis, MO). All mice used in experiments were male. Ages of mice are reported in figure legends or denoted in figure panels. For experiments investigating kidney histology and function mice were 16–24 weeks old. No animals were excluded from the studies performed. All animals were maintained on a normal chow diet and housed in a room with a 12:12-h light/dark cycle and an ambient temperature of 22 °C.

**Transgenic mice overexpressing miR-93.** The 309-bp miR-93 precursor was PCR amplified from genomic DNA isolated from mouse podocytes and introduced downstream of the GFP coding region using Xho1 to Not1 restriction sites into the LB2-FLIP-Ubic lentivirus vector. We cloned the expression cassette in an inverse, anti-sense orientation between two different pairs of loxP sites. The construct was designed so that Cre induction could be used to mediate the inversion of the GFP and pre-miR-93 cassette into a sense orientation[50]. Cre induction mediates an initial flipping, followed by deletion of selection markers, Puromycin and Thy1.1, based on orientation and location of two sets of mutant loxP sites. The expression cassette is under the control of the *Ubiquitin C* (*ubic*) promoter. Lentiviral particles for microinjection were generated by the Diabetes and Endocrinology Research Center (DERC) Vector Core at BCM (http://www.bcm.edu/research/centers/diabetes-research/core-labs/gene-vector-core site). Transgenic miR-93 (*miR-93*$^{Tg}$) mice were generated by lentiviral transduction of preimplantation C57/Bl6 blastocysts. Floxed *miR-93*$^{Tg}$ mice bred in expected Mendelian frequencies and proportional male/female ratios. Transgenic mice were generated by the Genetically Engineered Mouse Core at BCM (http://www.bcm.edu/research/advanced-technology-core-labs/lab-listing/genetically -engineered-mouse/home.htm) by infection of preimplantation C57/Bl6J blastocysts with the LB2-FLIP-miR-93$^{Tg}$ lentivirus. Founder tail DNA was genotyped with primers to detect GFP, Puromycin and Thy1.1 found in Supplementary Table 2. All progeny thereafter were genotyped using primers amplifying the GFP transgene region.

**Mimics, siRNA and target site protector reagents.** Pre-miRNA precursor molecules and anti-miR oligos, ON-Target *plus* siRNA against mouse Msk2 and

**Figure 6 | Targeting Msk2 *in vivo* prevents progression of DN.** (**a**) Left, representative western blot analysis for Msk2 levels in podocytes isolated from non-diabetic (db/m) ($n=4$), db/db+shCtl ($n=3$), and db/db+shMsk2 ($n=5$) mice treated with shMsk2. Right, densitometric quantification of western blots to quantify Msk2 levels. (**b**) ACR of non-diabetic (db/m), db/db+shCtl ($n=3$) and db/db+shMsk2 ($n=5$). (**c**) Representative images of TEM micrographs, PAS staining and WT1 staining from shMsk2-treated mice. Scale bars, 0.5 μM (first column) or 50 μM (second and third columns). (**d**–**h**) Quantification of (**d**) mesangial matrix expansion, (**e**) WT1+ cells, (**f**) GBM thickness, (**g**) Foot process density and (**h**) Foot process width in shMsk2-treated mice and controls. (**i**) Representative immunofluorescence micrographs of kidney sections from shMsk2-treated mice and controls from **a**, stained with anti-synaptopodin and anti-H3S10P. Sections were counterstained with DAPI. Merged images are meant to highlight podocyte-specific staining and insets are meant to highlight podocyte-specific localization of H3S10P. Scale bars, 50 μM. (**j**) qPCR analysis of podocyte-specific genes, Wt1, podocin and nephrin from podocytes isolated from shMsk2-treated mice and controls. All expression values were normalized to Gapdh internal controls. (**k**) Schematic describing mode of action and general design of LNA-modified long RNA gapmers (**l**) Top, western blot analysis of podocytes transfected with Msk2 or Scramble, negative control gapmers with antibodies directed against Msk2, Msk1 and actin. Bottom, study design used for the administration of Msk2 or Scramble, negative control gapmers in db/db mice. (**m**) Western blot analysis of kidney cortices from db/m and db/db mice administered gapmers directed against Msk2 or Scramble, negative control gapmers, all values normalized to actin ($n=5$ mice/group). (**n**) ACR of non-diabetic (db/m), db/db mice administered gapmers directed against Msk2 or Scramble, negative control gapmers ($n=5$ mice/group). (**o**) ACR of 24-week-old db/m;Msk2+/+ ($n=3$), db/m;Msk2$^{-/-}$ ($n=5$), db/db;Msk2+/+ ($n=6$), and db/db;Msk2$^{-/-}$ ($n=6$) mice. Data expressed as mean ± s.e.m. ns: no significance, *$P<0.05$, **$P<0.01$, ***$P<0.001$. One-way analysis of variance with Tukey's post test for multiple comparisons was used for groups of three or more.

scramble siRNA controls were purchased from Thermo Fisher Scientific (Life Technologies, Grand Island, NY, USA). Custom VEGF target site protector and a non-targeting target site protector with LNA modifications were designed and synthesized by Exiqon Inc. (Woburn, MA, USA). All oligonucleotides were used in cell culture experiments at a final concentration of 50 nM and transfected into podocytes using Lipofectamine 2000 Transfection reagent (LifeTech).

**miR mimic and LNA-Gapmer injections.** The mmu-miR-93 and non-targeting (Cel-67) in vivo mimics were obtained from Thermo Scientific. Lyophilized mimics were resuspended in sterile saline and were stored as aliquots in − 80 °C, before use and mixing with TurboFect in vivo transfection reagent (Life Technologies, Grand Island, NY, USA) for i.p. injections according to manufacturer's instructions. Diabetic db/db mice were allocated in a randomized and blinded manner into miR-93 mimic or NT control-treated groups. The 8-week-old db/db mice were administered 100 μg of mature miR-93 mimic or a NT mimic via i.p. injection, twice a week for 10 weeks. The mmu-Msk2 and scramble Gapmers were custom designed by Exiqon Inc. Gapmer oligos were LNA modified with a phosphothiorate backbone. Lyophilized gapmers were resuspended in sterile saline and were stored as aliquots in − 80 °C, before use for i.p. injections according to manufacturer's instructions. db/m and db/db mice were purchased from Jackson Labs. db/db mice were allocated in a randomized and blinded manner into Msk2 gapmer or scramble gapmer-treated groups. 8-week-old mice were administered 100 μg of Msk2 gapmer or scramble gapmer via i.p. injection, twice a week for 8 weeks.

**In vivo Msk2 knockdown.** Kidney-specific knockdown of Msk2 was achieved by delivering Msk2 shRNA (mouse Rps6ka4, 29mer shRNA constructs in an untagged retroviral vector) or a scramble control shRNA construct purchased from OriGene (Rockville, MD, USA) with the kidney in vivo transfection reagent (KIDNEY-targeted In Vivo Transfection Reagent; Altogen Biosystems, Las Vegas, NV, USA). Overall, 50 μg of endotoxin-free plasmid DNA/mouse was coupled to the reagent according to manufacturer's guidelines immediately before injections. Eight-week-old diabetic (db/db) mice were divided into two groups, shMsk2 or shScramble control, and received injections once a week i.p., until 16 weeks. Urine was collected at the end of the 16 weeks and mice were sacrificed for further histological and biochemical analysis.

**Human kidney biopsy analysis.** Clinical data from patients with DN at the time of renal biopsy are summarized in Supplementary Table 1. Human kidney biopsy samples were obtained from Dr Katalin Susztak (BioBank at the University of Pennsylvania, Perlman School of Medicine)[51–53]. Total cortex RNA was isolated formalin fixed, paraffin embedded (FFPE) sections using QuickExtract FFPE RNA Extraction Kit (Epicentre, Madison, WI, USA). Overall, 10 ng of total RNA was subjected to miRNA specific RT and qPCR using TaqMan miRNA Specific Gene Assays (LifeTech) on a StepOnePlus Real Time PCR System (LifeTech). Results are reported as relative expression values normalized to U6 snRNA levels.

**Biochemical, histological and morphometric studies.** Urinary albumin was measured using Albuwell M (Excocell, Philadelphia, PA, USA) according to manufacturer's instructions. For histological staining procedures 4 μm thick formalin fixed, paraffin embedded kidney sections were deparaffinized and dehydrated using Histo-Clear and a series of increasingly concentrated ethanol washes. Before deparaffinization, sections were further adhered to glass slides by incubating in a hybridization oven at 60 °C for 1 h. Sodium citrate (10 mM, pH 6.4) antigen retrieval using a microwave oven was performed (3 min high power, 17 min 20% power) for the experiments involving H3S10P. Tris-EDTA (10 mM Tris, 1 mM EDTA, 0.05% Tween-20, pH 9.0) antigen retrieval using a microwave oven was performed (3 min high power, 17 min 20% power) was performed experiments for Synaptopodin, Nephrin, WT1 and Desmin. Serum creatinine and blood pressure measurements were conducted by the Baylor College of Medicine Center for Comparative Medicine Pathology Core and the Baylor College of Medicine Mouse phenotyping Core, respectively. PAS staining was performed according to manufacturer's guidelines (Sigma Aldrich, St Louis, MO, USA). For histological and immunofluoresnce analysis, the investigators analyzing data were blinded to the group allocations. PAS, TEM and SEM data was examined by an independent pathologist, blinded to the experimental conditions. Quantification of histological parameters (Mesangial Matrix Expansion, GBM Thickening, foot process density and width, desmin staining) were performed using ImageJ v2.0 Software[27].

**Glomeruli and podocyte isolation.** Glomeruli were isolated using magnetic Dynabeads (Life Technologies, Grand Island, NY, USA)[27]. Following endothelial cell depletion with selection by biotin-labelled CD31; podocytes were isolated by positive selection with biotin-labelled Kirrel3 and podocalyxin antibodies (2.5 μg/antibody/mouse, R&D Systems, Minneapolis, MN, USA). Podocytes were then isolated from the total cell population using magnetic, streptavidin-labelled Dynabeads[54]. Primary podocytes were established by passaging glomerular outgrowths 5 days after initial plating on collagen coated tissue culture plates.

To induce recombination of miR-93Pod[Tg] primary podocytes, cells were serum deprived for 24 h and treated with 1 μM 4-OH Tamoxifen or vehicle control.

**DNase-Seq.** DNA-seq data were obtained from differentiated cultured podo-cytes[22]. Five million nucleiper experimental condition, were subjected to digestion by 30 U DNaseI (Roche, Basel, CH). Digested nuclei were treated with proteinase K and RNase A, and DNA was extracted using phenol:chloroform and ethanol precipitated overnight at − 20 °C. DNA was size selected by gel electrophoresis and purified using the Qiagen Gel Purification Kit (Qiagen, Valencia, CA, USA). Genomic DNA was isolated and subjected to digestion by 1 U DNase1, to act as input. Fifty nanograms of size selected DNA were submitted for library prep. Biological replicate libraries were prepared for downstream analysis. The Illumina compatible libraries were prepared using KAPA Library preparation kit as per the manufacturer's protocol (Illumina, Inc., San Diego, CA, USA). In brief, DNA was fragmented to a median size of 150 bp by sonication. Fragmented DNA ends were polished and 5′-phosphorylated. After addition of 3′-A to the ends, indexed Y-adapters were ligated and the samples were PCR amplified. The resulting DNA libraries were quantified and validated by qPCR, and sequenced on Illumina's HiSeq 2000 in a single-read format for 36 cycles. The resulting BCL files containing the sequence data were converted into '.fastq.gz' files and individual sample libraries were demultiplexed using CASAVA 1.8.2 with no mismatches.

**RNA-Seq.** RNA from mouse podocytes was isolated as indicated below. Biological replicate libraries were prepared for downstream analysis. The Illumina compatible libraries were prepared using Illumina's TruSeq RNA Sample Prep kit v2 as per the manufacturer's protocol. In brief, Poly-A RNA was enriched using Oligo-dT beads. Enriched Poly-A RNA was fragmented to a median size of 150 bp using chemical fragmentation and converted into double stranded complimentary DNA (cDNA). Ends of the double stranded cDNA were polished, 5′-phosphorylated and 3′-A tailed for ligation of the Y-shaped indexed adapters. Adapter ligated DNA fragments were PCR amplified, quantified and validated by qPCR, and sequenced on Illumina's HiSeq 2000 (Illumina Inc., San Diego, CA, USA) in a paired-end read format for 76 cycles. The libraries were sequenced on a version 3 TruSeq paired-end flowcell according to manufacturer's instructions at a cluster density between 750–1,000 K clusters mm$^{-2}$. The samples containing four libraries per lane were sequenced for paired end 75 nt with a 7 nt read for indexes using version 3 sequencing reagents. The resulting BCL files containing the sequence data were converted into '.fastq.gz' files & individual sample libraries were demultiplexed using CASAVA 1.8.2 with no mismatches.

**RNA isolation and qPCR.** Isolation of RNA from cultured cells and tissue was performed using Trizol Reagent (LifeTech). RNA lysates were further column purified on the PureLink RNA mini kit (LifeTech). DNase1-treated RNA was quantified using a Nanodrop 2000 and samples with a 280/260 ratio between 1.9 and 2.1 were used for downstream applications. mRNA was reverse transcribed into cDNA using the iScript Reverse Transcription Supermix (BioRad, Hercules, CA, USA). cDNA for miRNA expression assays were generated from total RNA using the TaqMan Reverse Transcriptase (RT) kit and miRNA specific primer mixes. cDNAs were subjected to gene specific amplification with primers and miRNA specific probes listed in Supplementary Table 2. qRT-PCR's were performed on StepOnePlus Real Time System (LifeTech, Garland Island, NY, USA) using 2 × Power Sybr Green Master Mix (LifeTech, Garland Island, NY, USA). GAPDH and U6 snRNA were used as internal controls for mRNA and miRNA assays, respectively. Primer sequences and miRNA TaqMan assay ID's are listed in Supplementary Table 1.

**Western blot analysis.** Briefly, 30 μg protein from each sample was resolved by SDS–PAGE on Tris-Glycine 4–20% gradient gels (BioRad, Hercules, CA, USA), unless otherwise stated, and transferred to nitrocellulose membranes. Histones were extracted using the Episeeker Histone Extraction Kit (Abcam Inc., Cambridge, UK). Membranes were probed with primary antibodies overnight at 4 °C in blocking buffer. Membranes were washed three times for 5 min with Tris-buffered saline and Tween-20 (TBST), incubated in either HRP-goat-anti-mouse (1/5,000, BioRad) or HRP-goat-anti-rabbit (1/5,000, BioRad) secondary antibodies for 1 h at room temp. Immunoreactive bands were visualized using Pierce ECL plus western blotting substrate (Thermo Scientific). Primary antibodies included: Msk1 (Bethyl Labs, A302-747A, 1/500), Msk2 (Bethyl Labs, A302-746A, 1/500), H3S10P (Millipore, 05-1336, 1/1,000), VEGF (RB-222-P, Thermo Scientific, 1/200), H3S10PK14Ac (Millipore, 07-081, 1/500), H3K9K14Ac (Cell Signaling, 9677, 1/500) and β-actin (A2228, Sigma, 1/5,000). Original uncropped westerns are provided in Supplementary Fig. 10. Western blot signals were quantified by standard densitometric analysis using NIH ImageJ version 2.0 program.

**Subcloning of the Msk2 3′-UTR and luciferase reporter assays.** The 3′-UTR of the mouse Msk2 gene (NM_019924) was amplified from immortalized mouse podocytes genomic DNA by PCR using HotMaster Taq DNA Polymerase (5PRIME), with respective primers (Supplementary Table 2). The PCR products were cloned between XhoI and EcoRI sites of luciferase reporter vector 3.1-luc,

kindly provided by Dr Ralph Nicholas (Dartmouth Medical School, Hanover, NH, USA). The putative miR-93 binding site (CACUUU) in mouse *Msk2* was mutated by oligonucleotide-directed PCR with primers listed in Supplementary Table 2. For experiments using 3.1-luc luciferase reporter constructs *in vitro*, $1.5 \times 10^5$ HEK293T cells were plated in 12-well plates. Overall, 1 µg of either 3.1-luc empty vector, 3.1- luc-Msk2-Wt-3UTR or 3.1-luc-Msk2-Mut-3UTR and 50 ng of pSV-β-gal control vector (Promega) and 50 nM of miR-93 mimics were transfected using Lipofectamine 2000 (Life Technologies, Grand Island, NY, USA). And 48 h after transfection, luciferase activity was measured using Steady-Glo Luciferase Assay System (Promega, Madison, WI, USA) on a FLUOstar Omega luminometer (BMG Labtech, Cary, NC, USA) as previously reported[6,10]. Levels were normalized β-gal internal controls.

**RNA immunoprecipitation.** Mouse podocytes were transfected with miR-93 mimics, anti-miR-93 mimics and corresponding non-targeting controls. Cells were ultraviolet crosslinked 48 h after transfection and subjected to Argonaute immunoprecipitation with a Pan-Argonaute antibody (2A8, 5 µg/i.p., Millipore) and RNA isolation was performed as described previously[55].

**Target site protection.** Kidney organ culture and tissue delivery of target site protector oligos were carried out as reported previously with slight modifications[10]. Briefly, kidneys from VEGF-LacZ mice were collected under sterile conditions. Kidney capsules were removed and kidney cortices dissected and cut into small pieces (~1 mm). Kidney cortex pieces were cultured using a roller bottle incubator (model 1,000; Robbins Scientific, Sunnyvale, CA, USA) at 37 °C, 5% $CO_2$, 20% $O_2$, 75% $N_2$ for 72 h in Dulbecco's modified Eagle's medium, containing 10% fetal bovine serum, 1× anti-anti (Gibco-LifeTech), under either normal glucose (5 mM D-glucose) or high glucose conditions (25 mM D-glucose). Overall, 10 µM of a VEGF target site protector or NT target site protector were delivered into the cultured kidney pieces using 4 mM of the Endo-Porter delivery system (Gene Tools). After 72 h, kidney pieces were fixed in 2% paraformaldehyde and 0.2% glutaraldehyde. X-gal staining (Millipore) was performed at 37 °C in 0.02% glutaraldehyde, 5 mM $K_3Fe(CN)_6$, 5 mM $K_4Fe(CN)6$, and 2 mM $MgCl_2$ as described previously[10]. Kidney pieces were then post fixed in 4% paraformaldehyde followed by paraffin embedding. Paraffin sections were dewaxed and mounted onto glass slides for image acquisition. All images were obtained on Nikon Eclipse 50i Light microscope (Nikon Inc, Tokyo, Japan).

**Scanning electron microscopy.** Samples were treated with a fixative containing 3% glutaraldehyde plus 2% paraformaldehyde in 0.1 M cacodylate buffer, pH 7.3, overnight at 4 °C. The samples were washed with 0.1 M cacodylate buffer, pH 7.3 for 3 × 10 min. The samples were then post fixed with 1% cacodylate buffered osmium tetroxide for 1 h, washed with 0.1 M cacodylate buffer for 3 × 10 min, then in distilled water, 2 × 5 min. The samples were then dehydrated with a graded series of increasing concentrations of ethanol for 5 min each. The samples were then transferred to graded series of increasing concentrations of hexamethyldisilazane (HMDS) for 5 min each and air dried overnight. Samples were mounted on to double-stick carbon tabs (Ted Pella. Inc., Redding, CA, USA), which have been previously mounted on to aluminium specimen mounts (Electron Microscopy Sciences, Ft. Washington, PA, USA). The samples were then coated under vacuum using a Balzer MED 010 evaporator (Technotrade, NH, Manchester, USA) with platinum alloy for a thickness of 25 nm, then immediately flash carbon coated under vacuum. The samples were transferred to a desiccator for examination at a later date. Samples were examined in a JSM-5910 scanning electron microscope (JEOL, USA, Inc., Peabody, MA, USA) at an accelerating voltage of 5 kV.

**Transmission electron microscopy.** Fixed samples were washed in 0.1 M cacodylate buffer, post fixed with 1% buffered osmium tetroxide for 1 h, and stained en bloc with aqueous 1% Millipore-filtered uranyl acetate. The samples were washed several times in water, then dehydrated in increasing concentrations of ethanol, infiltrated, and embedded in LX-112 medium. The samples were polymerized in a 60 °C. oven for about 3 days. Ultrathin sections were cut in a Leica Ultracut microtome (Leica, Deerfield, IL, USA), stained with uranyl acetate and lead citrate in a Leica EM Stainer, and examined in a JEM 1010 transmission electron microscope (JEOL, USA, Inc., Peabody, MA, USA) at an accelerating voltage of 80 kV. Digital images were obtained using AMT Imaging System (Advanced Microscopy Techniques Corp, Danvers, MA, USA).

**Immunofluorescence.** Podocytes were grown on 35 mm BioCoat Collagen I coated coverslips (BD, San Jose, CA, USA). After differentiation and treatment as indicated, cells were washed twice with PBS and fixed in 4% paraformaldehyde (PFA) for 10 min at room temperature. Cells were washed three times with PBS and permeabilized with 0.2% Triton X-100. Cells were washed twice with PBS and blocked in 10% donkey serum for 30 min at room temperature. The following primary antibodies Msk2 (R&D Systems, MAB2310, 1/200), H3S10P (Cell Signaling, 9701, 1/200), Synaptopodin (Santa Cruz Biotechnology, sc-21537). Nephrin (Fitzgerald, 20R-NP002, 1/100), WT1 (Santa Cruz Biotechnology, sc-192, 1/200). Podocytes were incubated in primary antibody, overnight at 4 °C, and then

washed three times with 1× PBS. Fluorophore conjugated secondary antibodies to rat (1/200) or rabbit (1/200) were applied to their respective primary antibodies and incubated at room temperature for 1 h in the dark. Cells were washed three times with PBS and nuclei were counterstained with DAPI (Thermo Scientific) and mounted with ProLong Gold Antifade mounting reagent (LifeTech, Garland Island, NY, USA). Images were obtained using the Deltavision Deconvolution microscope (Applied Precision, LLC, Issaquah, WA, USA). For quantification purposes, NIH ImageJ version 1.62 was used, fluorescent intensity was quantified in nuclei of cells with built in software. Kidneys for immunofluoresnce analyses were drop-fixed in 4% PFA overnight at 4 °C. Kidneys were embedded in Tissue-Tek O.C.T. compound (Sakura FineTek, Torrance, CA, USA) and frozen on dry ice and isopentane. Sections from Podocin-mTom/mGFP mice were imaged directly. Sections for immunofluoresence were processed as above. All kidney sections were counterstained with DAPI.

**RNA-Seq analysis.** The raw data quality was assessed using FastQC software. Adaptor presence was tested using Trimmomatic. Reads from each sample were aligned to the NCBI mouse reference genome build 37.2 using Tophat2 v2.0.4. Transcript quantification, normalization and assembly were carried out using Cufflinks[56]. Cuffdiff2 was used to identify features differentially expressed between conditions. A 0.05 false discovery rate was used in selecting significant genes. To further define differentially regulated genes, a cutoff of $-\log_{10}$ (P value) $>1$ and a $\log_2$ fold change (FC) of $<0.5$ or $>0.5$ was employed. TargetScan Mouse Release 5.2 was then run on input lists to find predicted miR-93 binding sites. Distributions of the relative expression levels of all genes containing 7-mer or 8-mer seeds were plotted to evaluate the effect of miR-93 overexpression on genes containing miR-93 seed sequences. Pathway analysis was conducted using GSEA[20,57]. Significantly downregulated genes were analyzed for motif enrichment using miRVestigator[58].

**DNase-Seq analysis.** DNase-seq reads underwent strict quality control processing with the TrimGalore! Package to remove low quality bases as well as adapter sequences. Reads that passed quality control were mapped to the mm10 version of the mouse genome using the Bowtie short read aligner[59]. Duplicate reads were then filtered. Reads were shifted 50 nt downstream of their mapped location and the number of complete reads that mapped within 500 nt of each unique transcription start site as defined by ENSEMBL vGRCm38.73 of the mouse transcriptome were used for hypersensitivity analysis[60]. TSSs that had at least the equivalent of 10 reads per million in a single condition, relative to the condition with the largest number of mapped reads, were kept for subsequent analysis. The number of DNase-seq reads per TSS were normalized across conditions to eliminate systematic differences by the DESeq algorithm[61]. A pooled dispersion estimation was then done in DESeq, followed by a negative binomial test to identify differences in enrichment of DNase-seq reads across each of the three conditions. Pathway analysis was carried out on the top 200 most differentially enriched TSSs in each of the conditions using the GSEA resources as before. Peaks were called using the MACS2 peak-calling algorithm using the parameters recommended by for DNase-seq data[62]. The input DNA was used as a control for each of the three samples. Differential DNAse-seq scores were calculated as described previously[25]. Sites with positive value scores were considered more hypersensitive, sites with negative value scores were considered less hypersensitive.

**Nephroseq analysis.** Human *MSK2* and *MCM7* expression data were downloaded from the Woroneckia *et al.*[51] and Ju *et al.*[63] datasets using the Nephroseq data mining platform (www.nephroseq.org).[19]

**Statistical analysis.** Group data are all expressed as mean ± s.e. Comparisons of multiple groups were performed using one-way analysis of variance (one-way ANOVA) followed by Tukey's multiple comparisons test. Comparisons between two groups were performed using the student's *t*-test. All tests were two-tailed, with a $P < 0.05$ considered to be statistically significant. Tests were performed with GraphPad version 6.0b (Graphpad Software Inc. La Jolla, CA, USA). We employed albuminuria data from db/db mice from previous experiments and published data[27] to estimate the number of mice required to achieve 80% power, with a 0.05 significance level.

**Data availability.** High-throughput sequencing data that support the findings of this study have been deposited in GEO with primary accession code GSE64081. The data that support the findings of this study are available from the corresponding author upon request.

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

## Acknowledgements

This work was supported by NIH Grants RO1DK091310 and RO1DK078900 (F.R.D), and T32GM088129 (S.S.B). Work performed by the UT-MDACC high-resolution microscopy facility was supported by Institutional funds (Core Grant CA16672). High-throughput sequencing data was generated by the UT-MDACC Sequencing and Microarray Facility (Core Grant CA016672 SMF). Analysis of RNA-Seq data was performed by the UT-MDACC Bioinformatics and Computational Biology Shared Resource (Mary K. Chapman Foundation, Support Grant CA016672). Analysis of DNase-Seq data was performed by the Duke University Integrative Genomic Shared Resource (http://www.genome.duke.edu/cores/analysis). Blood pressure measurements were performed by the Baylor College of Medicine Mouse Phenotyping Core. Lentivirus for transgenic mice was generated by the Diabetes and Endocrinology Research Center (P30DK079638) at Baylor College of Medicine. Mouse embryo infections were performed by the Genetically Engineered Mouse (GEM) Core at Baylor College of Medicine (http://www.bcm.edu/research/advanced-technology-core-labs/lab-listing/genetically-engineered-mouse/home.htm).

## Author contributions

S.S.B., B.H.C., D.L.C., P.O. and F.R.D. designed experiments and analyzed data. S.S.B. and F.R.D wrote the manuscript. S.S.B., Y.W. and J.L. performed experiments and analyzed the data.

## Additional information

**Competing financial interests:** The authors declare no competing financial interests.

