## [Peer review file · Nature Communications]

Reviewers' comments:

Reviewer #1 (Remarks to the Author):

The manuscript has been improved, and the author has responded to my previous questions in concerns in the last review. Although only a supplemental figure, I still have concerns about the data presented in supplemental figure 9 because of the small "n" and the large standard deviation. Although the trends appear consistent with the hypothesis, the results are not interpretable because of this large individual variation and I would not include these data in the manuscript unless "n" is increased.

Reviewer #3 (Remarks to the Author):

General comments

Diabetic nephropathy (DN) is a serious microvascular complication of diabetes and is the leading cause of end-stage renal disease (ESRD) in industrialized countries. Diabetic nephropathy is associated with changes in the architecture of podocytes foot processes and in the structure of the slit diaphragm. Progressive podocyte injury plays a central role in the development of DN in both type 1 and type 2 diabetes. Despite the crucial importance of the kidney, both as therapeutic target and as determinant of the prognosis of patients with DN, little is known about the mechanisms underlying kidney damage associated with diabetes. Therefore, it is important to define the cellular mechanisms that determine and maintain normal podocyte structure and function.

The authors previously identified mir-93 as a signature miRNA during DN. They found that mir-93 is downregulated in cultured podocytes after high glucose treatment and that mir-93 represses VEGF-A in normoglycemic conditions. In the present work, they characterized Msk2, a novel target of miR-93, as a key mediator of chromatin remodeling and a novel target in diabetic nephropathy. This is an original and novel study.

To further examine the role of Msk2 in DN in vivo, the authors generated diabetic db/db mice and found that Msk2 KO mice do not exhibit any overt phenotypes. However, under diabetic conditions, these mice exhibit a significant reduction in albuminuria.

In the revised version of their manuscript, Badal et al have provided further studies to bolster their evidence for an important role of miR-93 - Msk2 in the progression of DN. The authors went on by selectively depleting Msk2 in kidneys of diabetic (db/db) mice by using a nanoparticle-based liposomal reagent delivery system coupled with a shRNA plasmid against Msk2. Diabetic db/db mice receiving this targeted intervention had significantly reduced albuminuria and showed improved kidney histology compared to mice receiving a non-targeting shRNA plasmid, providing further evidence of the therapeutic value of targeting Msk2 in DN. Msk2 inhibition was associated with reduced phosphorylated status of the chromatin marker, H3S10.

These results strengthen the manuscript. There are still some issues that need to be addressed. The picture is still somewhat blurred by attempts to imply a prominent role of the miR-93-Msk2 axis cascade in podocytes during DN. This cascade is validated in podocytes but it is not clear that protection conferred by podocyte-specific overexpression of miR-93 and systemic MSK2 inhibition was achieved through the same mechanisms. Furthermore, this is barely discussed.

Specific comments to rebuttal from Reviewer #3:

Although a considerable amount of experiments with convincing computational biology has been carried out, this work is still incomplete in several aspects. It looks the authors got somewhat overwhelmed by interesting genomic analyses and neglected straightforward demonstration of a detailed cellular and molecular cascade

In fact the miR-93-Msk2-H3S10P pathway is convincingly and originally demonstrated but the impact of chronic miR-93 mimic administration and Msk2 depletion strategies is pleiotropic. The manuscript plays down this possibility from abstract to discussion section.

Response: We agree with the reviewer that the effects of miRNAs in general are pleiotropic. We have not made any attempt to downplay this feature of miRNAs. In fact, we emphasize that by modulating global chromatin remodeling, miR-93 is able to greatly expand its targetome leading to widespread changes that ultimately serve to prevent DN progression.

R3: Point taken, although the structure of the manuscript still overemphasize the role of the miR-93-Msk2 cascade in podocytes when too limited investigation of the impacts of miR-93 manipulation in podocytes did not demonstrate how podocyte alteration limited experimental DN.

Furthermore, such therapeutic strategies are not likely to be used to treat patients with incipient diabetic nephropathy for years. Direct systemic miR manipulation or MSK2 depletion are being developed for acute and sub-acute diseases. Again, this deserves discussion. Could Msk2 pharmacological inhibition be envisioned?

Response: We agree that based on our findings in diabetic mice, pharmacological targeting of Msk2 is a highly innovative approach. Indeed, intense studies are currently underway in our laboratory to address this important question. Regarding systemic miRNA manipulation, as we have used in this manuscript, several recent review articles have addressed the emerging role of systemic miR manipulations in vivo and in vitro. We have added these references and expanded our discussion to address these points in our manuscript.

R3: The Reviewer acknowledges the useful additional references and the significant effort studying in effect of Msk2 depletion in vivo by using multiple different approaches. These results markedly strengthen the manuscript.

Some important previous comments have not been addressed. Here follows a short list of major flaws that should be addressed in any revision:

1- Quality of the morphological evidence: Unfortunately, the technical quality of most of the figures is such that the conclusions drawn could be considered as over interpretations. In particular, the work displays a crucial lack of precise localization of glomerular miR-93, Msk2 and H3S10P mRNA. In-situ hybridization in mouse and human kidneys and/or RT-qPCR in sorted primary podocytes from mouse models would demonstrate the hypothesized miR-93 - Msk2 pathogenic cascade.

Response: We disagree with the reviewer that our morphological evidence is overinterpreted. In Situ hybridization of miR-93 has been previously published by our group (Long et al., JBC, 2010, 285:23457-23465). We have verified that there is podocyte specific overexpression of miR-93 in our tamoxifen inducible, transgenic miR-93 mice (Fig. 1f), and we confirmed that our podocyte-specific transgenic construct was expressing its GFP reporter (Supp Fig. 2f). We went on to show that diabetic miR-9PodTg mice treated with tamoxifen as well as miR-93 mimic treated mice exhibited reduced levels of podocyte-localized Msk2 and H3S10P proteins (Figs. 5h-k). Moreover, these findings have also been shown in cultured podocytes.

Taken together, we believe that these data sufficiently support our main conclusions on the role of miR-93-Msk2-H3S10P cascade in vivo and in vitro. Please also note that although we are focusing on the kidney and podocytes in this manuscript, we concede, and in fact strongly believe, that similar cascades exist in other cells/tissues.

R3: The reviewer is not questioning the validity of the artificial podocyte specific overexpression of miR-93 in the tamoxifen inducible, transgenic miR-93 mice (Fig. 1f) or of impact on podocyte-localized Msk2 and H3S10P proteins in such mice in diabetes condition (Figs. 5h-k) or localized Msk2 and H3S10P proteins in immortalized podocytes.

As pointed out by the authors, I also do refer to their previous paper (JBC 285, page 23460, 2010, Figure 2b) in which they provide in situ hybridization for the localization of miR-93 and show that this microRNA is massively expressed by tubuli. MSK2 is also ubiquitously expressed. Furthermore, DN also involves significant tubulo-interstitial changes and capillary rarefaction. Therefore this important aspect of the paper would need much better level of proof to show that podocyte phenotype modulation and DN limitation by miR93 transgenic expression involved Msk2 and H3S10P.

Furthermore, fair discussion of most of other non-podocyte specific data is required. i.e. Although the miR-93 PodTg model suggests that the cascade is relevant to podocyte during DN, more diffuse expression pattern in non podocyte cells (tubuli, endothelial cells) and efficacy of systemic approaches with miR93 mimic and MSK2 modulators strongly suggest a global implication in DN

progression.

If such thorough discussion is not carried out, then the authors should correctly demonstrate at every step that the studied cascade takes place in podocytes in vivo with isolation of glomeruli and podocyte sorting and that podocyte Msk2 inhibition is sufficient to slow down experimental DN. The authors were suggested to create a podocyte-specific Msk2 knockout mouse (mouse line carrying Msk2 floxed alleles is commercially available) to further support their findings. Alternatively, they may test the efficacy of Msk 2 inhibition in a model of podocyte-specific miR-93 gene inducible abrogation to make sure that Msk2 inhibition prevented DN through a podocyte specific miR93-dependent manner. It is not an issue per se to show pleiotropic actions of therapeutic strategies. It is a questionable to oversimplify a complex pathophysiological process to build a podocentric interpretation while ignoring the other evidence. This very interesting study may nicely use the podocyte as a model to study the miR-93-Msk2-H3S10P pathway with acknowledgement of limitations and existence of other likely cellular targets. The lack of documentation of MSK2 and H3S10P levels in human tissues is a also a limitation to firmer translational interpretation.

Statistics: The discussion about appropriate statistical tests is complex when it comes to very small samples. I acknowledge there are no objections to using a t-test with extremely small samples, as long as the effect size is large. A permutation test may be useful for analyzing data sampled from a highly skewed distribution. However, permutation tests or other resampling techniques, such as bootstrapping and jackknifing, do not overcome the weakness of small samples in statistical inference. It is the responsibility to the Journal to accept or not experiments with small samples.

In short, Badal et al. made a compelling case for Msk2 involvement in progression of DN. Meanwhile, the role of podocyte has not been clarified and put in perspective as pointed out by Reviewers #2 and #3. The manuscript should be remodeled accordingly or additional experiments carried out.

Reviewer #2 (Remarks to the Author):

While the authors have added new data that support their hypothesis, several important questions remain unanswered. These comprise a direct demonstration of the protective effect from DN damage by podocyte targeted genetic deletion of the components of their system.

Further on, the results obtained on FFPE sections of human kidneys may be meaningless, given the high concentration of miR-93 in tubuli vs podocytes.

Response: The levels of miR-93 are already significantly reduced in experimental models of DN, and the main goal of our manuscript was to test whether overexpression of miR-93 could rescue the diabetic phenotype in experimental models of DN. However, since we found that Msk2 serves as a novel target of miR-93 whose role was unknown in DN, we extensively examined the effect of Msk2 depletion in vivo by using multiple different approaches, including by adapting Msk2 knockout mice, LNA-Gapmers and shRNA directed against Msk2 (Fig. 6 and Supp. Fig 10). All suggest that Msk2 is a central target of miR-93 and targeting Msk2 in diabetic mice can prevent progression of DN.

Comment to Response: Correct, the authors made a compelling case for Msk2 involvement in progression of DN. Meanwhile, the role of podocyte has not been clarified and put in perspective as pointed out by the Reviewer. The manuscript should be remodeled accordingly.

Regarding our FFPE analysis, we disagree with the reviewer that this analysis is meaningless since it correlates with the changes in miR-93 expression observed in our experimental models, and thus it underscores the potential importance of miR-93 dysregulation in human subjects with DN. It is true that miR-93 is expressed both in glomeruli as well as in tubuli. However, these findings do not diminish the importance of targeting miR-93 in patients with DN since any future therapies directed against miR-93 most likely would not be cell-specific.

Comment to Response: It is correct that the work suggests the importance of targeting miR-93 in patients with DN. Again, no strong and convincing link has been made between the mechanisms

that are at play in transgenic Pod-miR93 mice and systemic interventions in diabetic mice. It is not exact to start the Abstract with words such as "How podocytes respond to metabolic cues in their environment remains a central question in kidney research... » or the Discussion section with "The molecular mechanisms linking metabolic changes in the cytoplasm to chromatin reorganization in podocytes remain poorly understood", because the present study brought very little information about podocyte biology. The use of transfections in immortalized podocytes and data from whole kidney cortices do not help grasping pathophysiological mechanism at the cellular level in vivo.

Reviewers' comments:

Reviewer #1 (Remarks to the Author):

The manuscript has been improved, and the author has responded to my previous questions in concerns in the last review. Although only a supplemental figure, I still have concerns about the data presented in supplemental figure 9 because of the small "n" and the large standard deviation. Although the trends appear consistent with the hypothesis, the results are not interpretable because of this large individual variation and I would not include these data in the manuscript unless "n" is increased.

We appreciate the reviewer's suggestion. In the revised version of our manuscript, we have removed Supplementary Fig. 9. We feel that removing this supplementary experiment does not change the main conclusions of our manuscript.

Reviewer #2 (Remarks to the Author):

Not applicable/No comments.

Reviewer #3 (Remarks to the Author):

General comments

Diabetic nephropathy (DN) is a serious microvascular complication of diabetes and is the leading cause of end-stage renal disease (ESRD) in industrialized countries. Diabetic nephropathy is associated with changes in the architecture of podocytes foot processes and in the structure of the slit diaphragm. Progressive podocyte injury plays a central role in the development of DN in both type 1 and type 2 diabetes. Despite the crucial importance of the kidney, both as therapeutic target and as determinant of the prognosis of patients with DN, little is known about the mechanisms underlying kidney damage associated with diabetes. Therefore, it is important to define the cellular mechanisms that determine and maintain normal podocyte structure and function. The authors previously identified miR-93 as a signature miRNA during DN. They found that miR-93 is downregulated in cultured podocytes after high glucose treatment and that miR-93 represses VEGF-A in normoglycemic conditions. In the present work, they characterized Msk2, a novel target of miR-93, as a key mediator of chromatin remodeling and a novel target in diabetic nephropathy. This is an original and novel study.

To further examine the role of Msk2 in DN in vivo, the authors generated diabetic db/db mice and found that Msk2 KO mice do not exhibit any overt phenotypes. However, under diabetic conditions, these mice exhibit a significant reduction in albuminuria. In the revised version of their manuscript, Badal et al have provided further studies to bolster their evidence for an important role of miR-93-Msk2 in the progression of DN. The authors went on by selectively depleting Msk2 in kidneys of diabetic (db/db) mice by using a nanoparticle-based liposomal reagent delivery system coupled with a shRNA plasmid against Msk2. Diabetic db/db mice receiving this targeted intervention had significantly reduced albuminuria and showed improved kidney histology compared to mice receiving a non-targeting shRNA plasmid, providing further evidence of the therapeutic value of targeting Msk2 in DN. Msk2 inhibition was associated with reduced phosphorylated status of the chromatin marker, H3S10.

These results strengthen the manuscript. There are still some issues that need to be addressed. The picture is still somewhat blurred by attempts to imply a prominent role of the miR-93-Msk2 axis cascade in podocytes during DN. This cascade is validated in podocytes but it is

not clear that protection conferred by podocyte-specific overexpression of miR-93 and systemic MSK2 inhibition was achieved through the same mechanisms. Furthermore, this is barely discussed.

Specific comments to rebuttal from Reviewer #3:

Although a considerable amount of experiments with convincing computational biology has been carried out, this work is still incomplete in several aspects. It looks the authors got somewhat overwhelmed by interesting genomic analyses and neglected straightforward demonstration of a detailed cellular and molecular cascade. In fact the miR-93-Msk2-H3S10P pathway is convincingly and originally demonstrated but the impact of chronic miR-93 mimic administration and Msk2 depletion strategies is pleiotropic. The manuscript plays down this possibility from abstract to discussion section.

Response: We agree with the reviewer that the effects of miRNAs in general are pleiotropic. We have not made any attempt to downplay this feature of miRNAs. In fact, we emphasize that by modulating global chromatin remodeling, miR-93 is able to greatly expand its targetome leading to widespread changes that ultimately serve to prevent DN progression.

R3: Point taken, although the structure of the manuscript still overemphasize the role of the miR- 93-Msk2 cascade in podocytes when too limited investigation of the impacts of miR-93 manipulation in podocytes did not demonstrate how podocyte alteration limited experimental DN.

We would like to thank the reviewer for the extensive and thorough analysis of our manuscript. We have modified our manuscript accordingly. We have emphasized that we used the podocytes as a model system to monitor changes related to DN. Although we specifically overexpressed miR-93 in podocytes, we have included in our discussion that the renoprotective effects of miR-93 and Msk2 could be mediated, in part, by their effect on other cells within the kidney.

Furthermore, such therapeutic strategies are not likely to be used to treat patients with incipient diabetic nephropathy for years. Direct systemic miR manipulation or MSK2 depletion are being developed for acute and sub-acute diseases. Again, this deserves discussion. Could Msk2 pharmacological inhibition be envisioned?

Response: We agree that based on our findings in diabetic mice, pharmacological targeting of Msk2 is a highly innovative approach. Indeed, intense studies are currently underway in our laboratory to address this important question. Regarding systemic miRNA manipulation, as we have used in this manuscript, several recent review articles have addressed the emerging role of systemic miR manipulations in vivo and in vitro. We have added these references and expanded our discussion to address these points in our manuscript.

R3: The Reviewer acknowledges the useful additional references and the significant effort studying in effect of Msk2 depletion in vivo by using multiple different approaches. These results markedly strengthen the manuscript.

We appreciate the reviewer's comment related to our strategies for understanding the role of Msk2 in DN.

Some important previous comments have not been addressed. Here follows a short list of major flaws that should be addressed in any revision: 1- Quality of the morphological evidence: Unfortunately, the technical quality of most of the figures is such that the conclusions drawn could be considered as over interpretations. In particular, the work displays a crucial lack of precise localization of glomerular miR-93, Msk2 and H3S10P mRNA. In-situ hybridization in mouse and human kidneys and/or RT-qPCR in sorted primary podocytes from mouse models would demonstrate the hypothesized miR-93 - Msk2 pathogenic cascade.

Response: We disagree with the reviewer that our morphological evidence is overinterpreted. In Situ hybridization of miR-93 has been previously published by our group (Long et al., JBC, 2010, 285:23457-23465). We have verified that there is podocyte specific overexpression of miR-93 in our tamoxifen inducible, transgenic miR-93 mice (Fig. 1f), and we confirmed that our podocytespecific transgenic construct was expressing its GFP reporter (Supp Fig. 2f). We went on to show that diabetic miR-9PodTg mice treated with tamoxifen as well as miR-93 mimic treated mice exhibited reduced levels of podocyte-localized Msk2 and H3S10P proteins (Figs. 5h-k). Moreover, these findings have also been shown in cultured podocytes. Taken together, we believe that these data sufficiently support our main conclusions on the role of miR-93-Msk2-H3S10P cascade in vivo and in vitro. Please also note that although we are focusing on the kidney and podocytes in this manuscript, we concede, and in fact strongly believe, that similar cascades exist in other cells/tissues.

R3: The reviewer is not questioning the validity of the artificial podocyte specific overexpression of miR-93 in the tamoxifen inducible, transgenic miR-93 mice (Fig. 1f) or of impact on podocytelocalized Msk2 and H3S10P proteins in such mice in diabetes condition (Figs. 5h-k) or localized Msk2 and H3S10P proteins in immortalized podocytes. As pointed out by the authors, I also do refer to their previous paper (JBC 285, page 23460, 2010, Figure 2b) in which they provide in situ hybridization for the localization of miR-93 and show that this microRNA is massively expressed by tubuli. MSK2 is also ubiquitously expressed. Furthermore, DN also involves significant tubulo-interstitial changes and capillary rarefaction. Therefore this important aspect of the paper would need much better level of proof to show that podocyte phenotype modulation and DN limitation by miR93 transgenic expression involved Msk2 and H3S10P.

Furthermore, fair discussion of most of other non-podocyte specific data is required. i.e. Although the miR-93 PodTg model suggests that the cascade is relevant to podocyte during DN, more diffuse expression pattern in non podocyte cells (tubuli, endothelial cells) and efficacy of systemic approaches with miR93 mimic and MSK2 modulators strongly suggest a global implication in DN progression.

If such thorough discussion is not carried out, then the authors should correctly demonstrate at every step that the studied cascade takes place in podocytes in vivo with isolation of glomeruli and podocyte sorting and that podocyte Msk2 inhibition is sufficient to slow down experimental DN. The authors were suggested to create a podocyte-specific Msk2 knockout mouse (mouse line carrying Msk2 floxed alleles is commercially available) to further support their findings. Alternatively, they may test the efficacy of Msk 2 inhibition in a model of podocyte-specific miR- 93 gene inducible abrogation to make sure that Msk2 inhibition prevented DN through a podocyte specific miR93-dependent manner. It is not an issue per se to show pleiotropic actions of therapeutic strategies. It is a questionable to oversimplify a complex pathophysiological process to build a podocentric interpretation while ignoring the

other evidence. This very interesting study may nicely use the podocyte as a model to study the miR-93-Msk2-H3S10P pathway with acknowledgement of limitations and existence of other likely cellular targets. The lack of documentation of MSK2 and H3S10P levels in human tissues is also a limitation to firmer translational interpretation.

We agree that targeted, podocyte-specific deletion of Msk2 in a model of DN would be another interesting approach to implicate miR-93/Msk2/H3S10P in DN. We have taken the reviewer and editors' advice and have modified our manuscript to address their main concern that we have overemphasized the podocyte-specific nature of our findings. To this end, we have modified the discussion emphasizing that while we observe renoprotection upon miR-93 overexpression in podocytes, our miR-93 mimic and Msk2 strategies suggest that, in addition to podocytes, other cell types may also be involved in mediating this renoprotective effect. This is due in large part to the broad distribution of miR-93 and Msk2, a statement that we have included in our discussion section. We recognize that future studies are necessary to further elucidate the tissue-specific impact of targeting Msk2 in DN (Pages 13 and 14).

Statistics: The discussion about appropriate statistical tests is complex when it comes to very small samples. I acknowledge there are no objections to using a t-test with extremely small samples, as long as the effect size is large. A permutation test may be useful for analyzing data sampled from a highly skewed distribution. However, permutation tests or other resampling techniques, such as bootstrapping and jackknifing, do not overcome the weakness of small samples in statistical inference. It is the responsibility to the Journal to accept or not experiments with small samples.

We would like to emphasize that similar sample sizes and statistical tests are routinely employed in comparable studies.

In short, Badal et al. made a compelling case for Msk2 involvement in progression of DN. Meanwhile, the role of podocyte has not been clarified and put in perspective as pointed out by Reviewers #2 and #3. The manuscript should be remodeled accordingly or additional experiments carried out.

Reviewer 2 - absent - mediated by reviewer 3:

Reviewer #2 (Remarks to the Author): While the authors have added new data that support their hypothesis, several important questions remain unanswered. These comprise a direct demonstration of the protective effect from DN damage by podocyte targeted genetic deletion of the components of their system.

Further on, the results obtained on FFPE sections of human kidneys may be meaningless, given the high concentration of miR-93 in tubuli vs podocytes.

Response: The levels of miR-93 are already significantly reduced in experimental models of DN, and the main goal of our manuscript was to test whether overexpression of miR-93 could rescue the diabetic phenotype in experimental models of DN. However, since we found that Msk2 serves as a novel target of miR-93 whose role was unknown in DN, we extensively examined the effect of Msk2 depletion in vivo by using multiple different approaches, including by adapting Msk2 knockout mice, LNA-Gapmers and shRNA directed against Msk2 (Fig. 6 and Supp. Fig 10). All suggest that Msk2 is a central target of miR-93 and targeting Msk2 in diabetic mice can prevent progression of DN.

Comment to Response: Correct, the authors made a compelling case for Msk2 involvement in progression of DN. Meanwhile, the role of podocyte has not been clarified and put in perspective as pointed out by the Reviewer. The manuscript should be remodeled accordingly.

We have addressed these concerns as detailed to reviewer 3 in the text above.

Regarding our FFPE analysis, we disagree with the reviewer that this analysis is meaningless since it correlates with the changes in miR-93 expression observed in our experimental models, and thus it underscores the potential importance of miR-93 dysregulation in human subjects with DN. It is true that miR-93 is expressed both in glomeruli as well as in tubuli. However, these findings do not diminish the importance of targeting miR-93 in patients with DN since any future therapies directed against miR-93 most likely would not be cell-specific.

Comment to Response: It is correct that the work suggests the importance of targeting miR-93 in patients with DN. Again, no strong and convincing link has been made between the mechanisms that are at play in transgenic Pod-miR93 mice and systemic interventions in diabetic mice. It is not exact to start the Abstract with words such as "How podocytes respond to metabolic cues in their environment remains a central question in kidney research... ? or the Discussion section with "The molecular mechanisms linking metabolic changes in the cytoplasm to chromatin reorganization in podocytes remain poorly understood", because the present study brought very little information about podocyte biology. The use of transfections in immortalized podocytes and data from whole kidney cortices do not help grasping pathophysiological mechanism at the cellular level in vivo.

We have modified the manuscript to include that our conclusions are not restricted to podocytes. The main sections modified are the Abstract and Discussion as suggested by the Reviewer. We do not believe that the Results Section warranted significant changes since we simply described our results related to podocytes as appropriate. In our discussion, we have mentioned that we assessed the value and /or limitations of targeting miR-93/Msk2/H3S10P pathway in the kidney. We recognize that in addition to podocytes, other cell types in the kidney could also be affected by miR-93 overexpression and Msk2 depletion.

REVIEWERS' COMMENTS:

Reviewer #3 (Remarks to the Author):

The authors have brought together a large amount of careful work that strongly suggests that targeting miR-93 and Msk2 in DN may represent a potential novel therapy. The revised version of the manuscript, although not fully discussing the limitations of the study in term of cellular insight, has taken my comments into consideration satisfactorily. The authors should be congratulated for such a novel approach.